# Associative Memories via Predictive Coding

**Tommaso Salvatori**[1], **Yuhang Song**[1,3,*], **Yujian Hong**[1], **Lei Sha**[1], **Simon Frieder**[1],
**Zhenghua Xu**[2], **Rafal Bogacz**[3], **Thomas Lukasiewicz**[1]
[1]Department of Computer Science, University of Oxford, UK
[2]State Key Laboratory of Reliability and Intelligence of Electrical Equipment,
Hebei University of Technology, Tianjin, China
[3]MRC Brain Network Dynamics Unit, University of Oxford, UK
{tommaso.salvatori, yuhang.song, yujian.hong, lei.sha, frieder.simon, thomas.lukasiewicz}
@cs.ox.ac.uk, zhenghua.xu@hebut.edu.cn, rafal.bogacz@ndcn.ox.ac.uk

## Abstract

Associative memories in the brain receive and store patterns of activity registered
by the sensory neurons, and are able to retrieve them when necessary. Due to their
importance in human intelligence, computational models of associative memories
have been developed for several decades now. In this paper, we present a novel
neural model for realizing associative memories, which is based on a hierarchical
generative network that receives external stimuli via sensory neurons. It is trained
using predictive coding, an error-based learning algorithm inspired by information
processing in the cortex. To test the model's capabilities, we perform multiple
retrieval experiments from both corrupted and incomplete data points. In an exten-
sive comparison, we show that this new model outperforms in retrieval accuracy
and robustness popular associative memory models, such as autoencoders trained
via backpropagation, and modern Hopfield networks. In particular, in completing
partial data points, our model achieves remarkable results on natural image datasets,
such as ImageNet, with a surprisingly high accuracy, even when only a tiny fraction
of pixels of the original images is presented. Our model provides a plausible
framework to study learning and retrieval of memories in the brain, as it closely
mimics the behavior of the hippocampus as a memory index and generative model.

## 1 Introduction

Throughout our lives, we learn a huge number of associations between concepts: the taste of a
particular food, the meaning of a gesture, or to stop when we see a red light. Every time we acquire
new information of this kind, it gets stored in our long-term memory, situated in distributed networks
of brain areas [1]. In particular, visual memories are stored in a hierarchical network of visual and
associative areas [2]. These regions learn progressively more abstract representations of visual stimuli,
so they participate in both perception and memory as each area memorizes relationships present in
their inputs [3]. Accordingly, early visual areas learn common regularities present in the stimuli [4],
while at the top of this hierarchy, associative areas (such as hippocampus, entorhinal cortex, and
perirhinal cortex) store the relationships between extracted features, which encode an entire stimulus
or episode [5]. The memory system of the brain is able to both recall complex memories [1, 6], and
use them to generate predictions to guide behavior [7]. Learning in these associative memories shapes
our understanding of the world around us and builds the foundations of human intelligence.

Building models that are able to store and retrieve information has been an important direction of
research in artificial intelligence. Particularly, such models include (auto)associative memories (AMs),
which allow for the storage of data points and their contents-based retrieval, i.e., for retrieving a stored

---

*Corresponding author

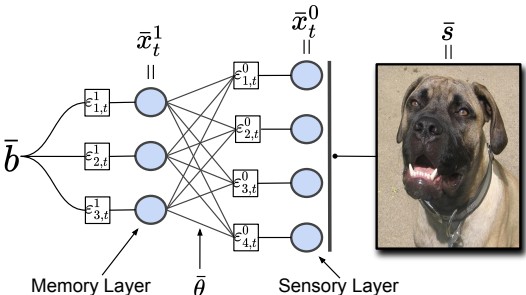

Figure 1: Generative PCN with two layers. Each line linking neurons denotes a pair of connections (excitatory in the left direction, and inhibitory in the right direction). At $t = 0$, the value nodes of the output layer are fixed to the vector representing the pixels of the figure. Then, during the learning phase, both weight parameters and value nodes are updated.

data point $s$ from a corrupted or a partial variant of $s$. One way to realize AMs is to store data points as attractors, so that they can be easily recovered via an energy minimization process when presenting their corrupted variants [8, 9]. Existing AMs include Hopfield networks [8] and modern Hopfield networks (MHNs) [10]. The latter are one-shot learners, which are able to store exponentially many memories, and to perfectly retrieve them. However, the retrieval process often fails when dealing with complex data, such as natural images. Recent works have shown that overparametrized autoencoders (AEs) are excellent AMs as well. Particularly, when training an AE to generate a specific point $s$ when $s$ itself is presented as an input, it gets stored as an attractor [11].

In this work, we present a novel AM model that is based on an energy-based generative approach. This AM model differs from Hopfield networks, as it is trained using predictive coding (PC), which is a biologically plausible learning algorithm inspired by learning in the visual cortex [4]. The idea that PC may naturally be related to AMs is inspired by recent works showing that the generative neural architecture that connects the hippocampus to the neocortex is based on an error-driven learning algorithm, which can be interpreted with a PC framework [6, 12]. From a machine learning perspective, predictive coding networks (PCNs) are able to perform both supervised and unsupervised tasks with a high accuracy [4, 13], and are completely equivalent to backpropagation when trained with a specific algorithm [14–16]. We show that the new AM model is not only interesting from a neuroscience perspective, but it also outperforms popular AM models when it comes to the storage and retrieval of complex data points. Our results can be briefly summarized as follows:

- We define generative PCNs and empirically show that they store training data points as attractors of their dynamics by demonstrating that they can restore original data points from corrupted versions. In an extensive comparison of the new AM model against standard AEs, the new model considerably outperforms AEs (in storage capacity, retrieval accuracy, and robustness) when tested on neural networks of the same size.

- The reconstruction of incomplete data points is a challenging task for AMs. Our model naturally solves the task of reconstructing complex and colored images with a surprisingly high accuracy. We also test our model on ImageNet, perfectly reconstructing single pictures even after removing all but $1/8$ of the original image. We then show that, to increase the overall capacity and retrieval robustness of the model, it suffices to add additional layers. We also compare our model against MHNs, showing that it significantly outperforms them in the aforementioned tasks.

## 2   Generative predictive coding networks

We now briefly recall predictive coding networks (PCNs), and we introduce generative PCNs, which are the underlying neural model for the novel AMs introduced in the subsequent section.

Deep neural networks have a multi-layer structure, where each layer is formed by a vector of neurons [17]. While in standard deep learning the goal is to minimize the error on a specific layer, PC defines an error in every layer of the network, minimized by gradient descent on a global energy function [4]. Particularly, let $M$ be a PCN with $L - 1$ fully connected layers of dimension $n$, followed by a fully connected layer of dimension $d$. We call the $d$-dimensional layer *sensory* layer (indexed as layer 0), which biologically corresponds to sensory neurons (see Fig. 1). We call the most internal layer (layer $L$) *memory*, which is equipped with an $n$-dimensional memory vector $b$. Every layer $l$ contains value nodes $x_{i,t}^l$, and every pair of consecutive layers is connected via weight matrices $\bar{\theta}^l$, which represent the synaptic weights between neurons of different layers. The value nodes, the

**Algorithm 1:** Learning to generate $\bar{s}$ with IL

---

**Require:** $\bar{x}_0$ is fixed to $\bar{s}$.

1: **for** $t = 0$ to $T$ **do**
2:     **for** each neuron $i$ in each level $l$ **do**
3:         update $x_{i,t}^l$ to minimize $E_t$ via Eq.(3)
4:     **end for**
5: **end for**;
6: update each $\theta_{i,j}^{l+1}$ and $b_i$ to minimize $E_t$ via Eqs. (4) and (5).

---

weight matrices, and the memory vector are all trainable parameters of the model. The signal passed from layer $l + 1$ to layer $l$, called prediction $\bar{\mu}_t^l$, is computed as follows:

$$\mu_{i,t}^l = \begin{cases} \sum_{j=1}^{n^{l+1}} \theta_{i,j}^{l+1} f(x_{j,t}^{l+1}) & \text{if } 0 \leq l < L \\ b & \text{if } l = L, \end{cases} \tag{1}$$

where $f$ is a non-linear activation function. To conclude, the difference between the value $\bar{x}_t^l$ and their predictions $\bar{\mu}_t^l$ is the *error* $\varepsilon_{i,t}^l = x_{i,t}^l - \mu_{i,t}^l$. We now describe how PCNs are trained. To do this, we explain one iteration of a training algorithm, called *inference learning* (*IL*), that is divided into an inference phase and a weight update phase.

*Inference:* Only the value nodes of the network are updated, while both the weight parameters and the memory vector are fixed. Particularly, the value nodes are modified via gradient descent to minimize the global error of the network, expressed by the following energy function $E_t$:

$$E_t = \tfrac{1}{2} \sum_{i,l} (\varepsilon_{i,t}^l)^2. \tag{2}$$

Assume that we train a generative PCN on a training point $\bar{s} \in \mathbb{R}^d$. To do this, the value nodes of the sensory layer are fixed to the training point $\bar{s}$, and are never updated. Thus, the error on every neuron of the sensory layer is equal to $\varepsilon_{i,t}^0 = s_i - \mu_{i,t}^0$. The process of minimizing $E_t$ by modifying all $x_{i,t}^l$ leads to the following changes in the value nodes:

$$\Delta x_{i,t}^l = \begin{cases} \gamma \cdot (-\varepsilon_{i,t}^l + f'(x_{i,t}^l) \sum_{k=1}^{n^{l-1}} \varepsilon_{k,t}^{l-1} \theta_{k,i,t}^l) & \text{if } 0 < l \leq L \\ 0 & \text{if } l = 0, \end{cases} \tag{3}$$

where $\gamma$ is the *integration step*, which is a constant determining by how much the activity changes in each iteration. The computations in Eqs. (1) and (3) are biologically plausible, as they have a neural implementation that can be realized in a network with value nodes $x_{i,t}^l$ and error nodes $\varepsilon_{i,t}^l$ [4], as shown in Fig. 1. The inference phase works as follows: starting from a given configuration of the value nodes $\bar{x}_0$, inference continuously upates the value nodes according to Eq. (3) until it has converged. We call the configuration of the value nodes at convergence $\bar{x}_T$, where $T$ is the number of steps needed to reach convergence (in practice, it is a fixed large number).

*Weight Update:* When the value nodes of the sensory layer are fixed to an input signal $\bar{s}$, inference may not be sufficient to reduce the total energy to zero. Hence, to further decrease the total error, a single *weight update* is performed: both the weight matrices and the memory vector are updated by gradient descent to minimize the same objective function $E_t$, and behave according to the following equations, where $\alpha$ is the learning rate. Particularly, the derived update rule is the following:

$$\Delta \theta_{i,j}^{l+1} = -\alpha \cdot \partial E_T / \partial \theta_{i,j}^{l+1} = \alpha \cdot \varepsilon_{i,T}^l f(x_{j,T}^{l+1}), \tag{4}$$

$$\Delta b_i = -\alpha \cdot \partial E_T / \partial b_i = -\alpha \varepsilon_{i,T}^L. \tag{5}$$

The phases of inference and weight update are iterated until the total energy $E_t$ reaches a minimum. This algorithm learns a dataset by using only local computations, which minimize the same energy function. Fig. 1 gives a graphical representation of generative PCNs, while the pseudocode is shown in Alg. 1. Detailed derivations of Eqs. (3) and (5) are in the supplementary material (and in [13]).

## 3 Predictive coding for associative memories

So far, we have shown how PCNs can perform generative tasks. We now show how generative PCNs can be used as associative memories, i.e., how the model stores the data points that it is trained on, and how these data points can be retrieved when presenting corrupted versions to the network, returning

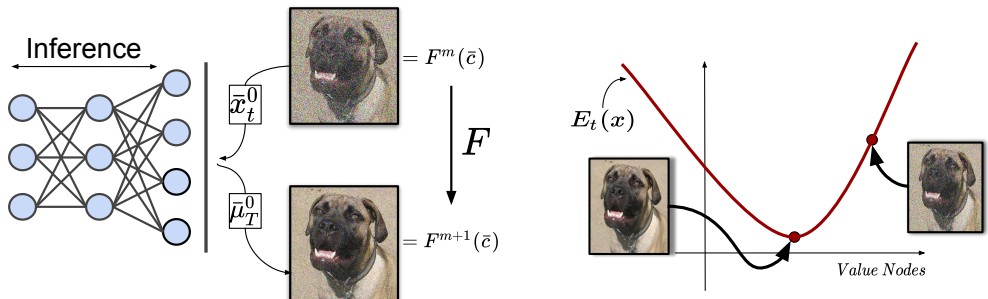

Figure 2: Left: Representation of the function $F$, used to retrieve stored images. It can be decomposed into three steps: (1) The value nodes of the sensory layer are fixed to the pixels of the corrupted image $F^m(\bar{c})$. (2) Inference runs for $T$ operations (until convergence). (3) We set $F^{m+1}(\bar{c})$ to the prediction of the sensory layer $\bar{\mu}_T^0$. Note that the weight parameters are never updated during the above steps. We have omitted the error nodes for simplicity. Right: Representation of an AM, where an image $\bar{s}$ (photo of a dog) is stored as an attractor of the dynamics. A corrupted image that lies in a specific neighborhood of $\bar{s}$ converges to it when minimizing the total energy via running inference.

the most similar stored data point. Let $\bar{s}$ be a training data point, and $M$ be the PCN considered above, already trained until convergence to generate $\bar{s}$. Moreover, assume that $M$ makes the total energy converge to *zero* at iteration $T$. At this point, the energy function defined on the value nodes has a local minimum $\bar{x}$ in which the value nodes of the sensory layer are equal to the entries of $\bar{s}$. Note that $\bar{x}$ is actually an attractor of the dynamics of $E_t$: when given a configuration that is not a local minimum, inference will update the value nodes until the total energy reaches a minimum. If this configuration lies in a specific neighborhood of $\bar{x}$, inference will converge to $\bar{x}$. So, given a dataset, we obtain an AM of the dataset if all the training points are stored in the above way as attractors.

The above can be used to retrieve stored data points $\bar{s}$: given a corrupted version $\bar{c} \in \mathbb{R}^d$ of $\bar{s}$, one can retrieve $\bar{s}$ as follows. First, we set the value nodes of the sensory layer to the corrupted points, i.e., $\bar{x}_t^0 = \bar{c}$ for the whole process. Then, we run inference until convergence and save the prediction $\bar{\mu}_T^0$ of the sensory layer. If the original data point was stored as an attractor, we expect the prediction $\bar{\mu}_T^0$ to be a less corrupted version of it. Let $F \colon \mathbb{R}^d \to \mathbb{R}^d$ be the function that sends $\bar{c}$ to $\bar{\mu}_T^0$ just described, and summarized in Fig. 2. Many iterations of this function allow to retrieve the stored data point. Hence, summarizing the above, training points are stored in the memory vector $\bar{b}_T$ and the weight parameters, and what the algorithm does to retrieve them is simply the inference phase of PCNs. Since visual memories are stored in hierarchical networks of brain areas, PC could be a highly plausible algorithm to better understand how memory and prediction work in the brain.

To experimentally show that generative PCNs are AMs, we trained a 2-layer network with ReLU non-linearity on a subset of 100 images of Tiny ImageNet and CIFAR10. After training, we presented the model with a corrupted variant (by adding Gaussian noise) of the training set. We then used the PCNs to reconstruct the original images from the corrupted ones. The experimental results confirm that the model is able to retrieve the original image, given a corrupted one. The obtained reconstructions for the Tiny Imagenet dataset (the most complex one, as each data point consists of $3 \times 64 \times 64$ pixels) are shown in Fig. 4. We now provide a more comprehensive analysis, which studies the capacity of generative PCNs when changing the number of data points and parameters.

*Experiments:* We trained 2-layer PCNs with ReLU non-linearity and hidden dimension $n \in \{512, 1024, 2048\}$ on subsets of the aforementioned datasets of cardinality $N = \{100, 250, 500, 1000\}$. Every model is trained until convergence, and all the images are retrieved as described in Section 3. To provide a numerical evaluation, an image is considered recovered when the mean squared error between the original image and the recovered image is less than $0.005$.

To compare our results against a standard baseline, we also trained 3-layer autoencoders (AEs) with the same hidden dimension on the same task, and compared the results. Note that the number of parameters of a 2-layer PCN is smaller than the one of a 3-layer AE with the same hidden dimension. This follows, as the additional layer (input layer, not needed in generative PCNs) almost doubles the number of parameters in some cases. Further details about the experiments and used hyperparameters are given in the supplementary material.

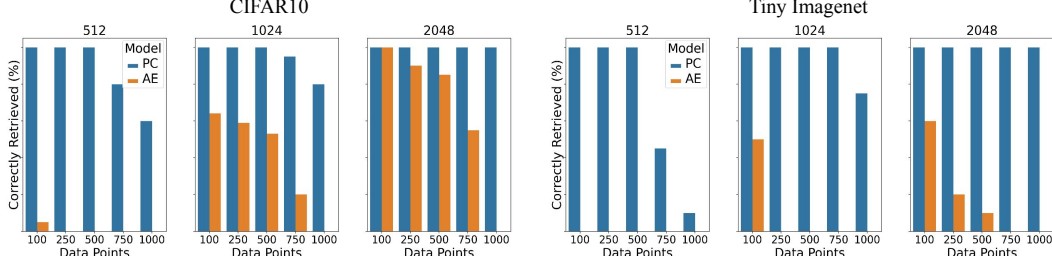

Figure 3: Percentage of correctly retrieved images by 2-layer generative PCNs (PC) and 3-layer autoencoders (AE) with hidden-layer dimensions of 512, 1024, and 2048, when presented with a corrupted image with Gaussian noise of variance $0.2$ of CIFAR10 and of Tiny ImageNet.

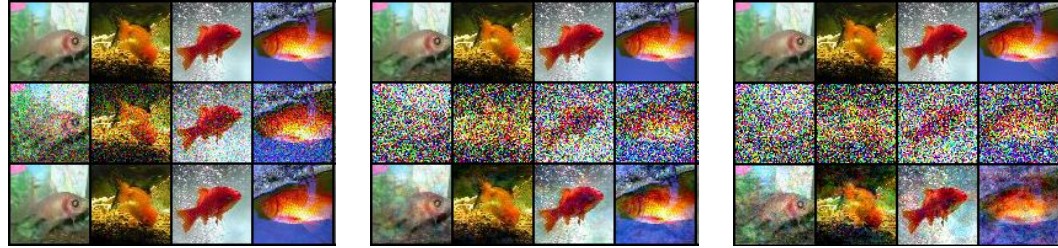

Figure 4: Reconstructions of $64 \times 64$ Tiny ImageNet images with noise levels $0.2$, $0.7$, and $0.9$, respectively (original, noisy, and reconstructed image in first, second, and third row, respectively).

*Results*: The analysis shows that our model is able to store and retrieve data points even when the network is not overparametrized (see Fig. 3). AEs trained with BP did not perform well: AEs with less than $1024$ hidden neurons always failed to restore even a single data point on the Tiny Imagenet dataset, and very few on other ones. The performance of AEs with $2048$ hidden neurons were always worse than PCNs with $512$ hidden neurons. This shows that overparametrization is essential for AEs to perform AM tasks, and that our proposed method offers a much more network-efficient alternative.

In terms of capacity, 2-layer PCNs with $512$ hidden neurons correctly store datasets of $250$ images of both CIFAR10 and Tiny ImageNet, and networks with $2048$ hidden neurons always store and retrieve all the presented datasets. As typical for AMs, small models trained on large datasets fail to store data points, as the space of parameters is not large enough to store each data point as an independent attractor. Our model is no different: PCNs with $512$ hidden units are able to store almost $500$ Tiny ImageNet images when trained on datasets of that size, but fail to store more than $200$ when trained on larger ones.

## 4 Retrieval from partial data points

So far, we have shown how the proposed generative model can be used to retrieve stored data points when presented with corrupted variants. We now tackle the different task of retrieving data points when presented with partial ones. Let $\bar{s}$ be the stored data point, and assume that a fraction of pixels of $\bar{s}$ are accessible, and the goal is to retrieve the remaining ones. Let $\bar{s}'$ be the vector of the same dimension of $\bar{s}$, where a fraction of pixels are equal to the ones of the stored data point, and assume that the position of the pixels that are equal to the ones of $\bar{s}$ is known. We now show how to retrieve the complete data point by using the same network $M$, trained as already shown in Section 3.

Let $M$ be a PCN trained to generate $\bar{s}$. Then, given the partial version of the original data point $\bar{s}'$, it is possible to retrieve the full data point using $M$ as follows: first, we only fix the value nodes of the sensory layer $\bar{x}_t^0$ to the entries of the partial data point $\bar{s}'$ that we know are equal to the ones of the stored data point, leaving the rest free to be updated. Then, we run inference on the whole network until convergence. At this point, we expect the value nodes $\bar{x}_T^0$ to have converged to the entries of the original data point, stored as an attractor. A graphical representation of the above mechanism is described in Fig. 5. To show the capabilities of this network, we have performed multiple experiments on the Tiny ImageNet and ImageNet datasets, and compared against existing models in the literature.

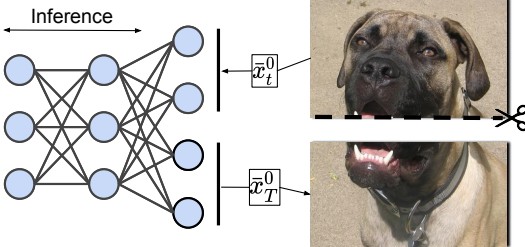

Figure 5: Algorithm used to retrieve a stored data point from a fraction of it: (1) The value nodes of the sensory layer that correspond to the available entries of $\bar{s}$ are fixed to the respective values. (2) Inference runs for $T$ operations (until convergence). (3) We set our prediction to be equal to the value nodes of the sensory layer $\bar{x}^0_T$. The weight parameters are never updated during the above steps. The error nodes are omitted for simplicity.

---

**Algorithm 2:** Retrieving $\bar{s}$ given a non-corrupted fraction $\bar{s}'$

---

**Require:** if $s'_i$ a correct entry of the original memory, then $x^0_{i,t}$ is fixed to $s'_i$.
1: **for** $t = 1$ to $T$ **do**
2:     **for** each neuron $i$ in each level $l$ **do**
3:         update $x^l_{i,t}$ to minimize $E_t$
4:     **end for**
5: **end for**;
6: **return** $\bar{x}^0_T$.

---

We now start by providing visual evidence on the effectiveness of this method. Note that the geometry of the mask does not influence the final performance, as our model simply memorizes single pixels.

*Experiments:* We trained two networks with hidden dimensions of $1024$ and $2048$, to generate $50$ images of the first class of Tiny ImageNet (corresponding to goldfishes), and a network of $8192$ hidden neurons to reconstruct $25$ ImageNet images. Then, we used inference as explained to retrieve the original images. We considered an image to be correctly reconstructed when the error between the original and the retrieved image was smaller than $0.001$. Furthermore, we plotted the partial images together with their reconstructions, for a visual check. Note that we used the thresholds that provided the fairest comparison: the denoising experiments fail to have a perfect retrieval, despite the fact that most of the images look visually good. Hence, we determined the threshold to be equal to $0.005$. Then, with the same threshold for the retrieval of partial images, our method always successfully retrieved all the images, and so we opted for a smaller threshold, which was more informative.

*Results:* On Tiny ImageNet, a generative PCN with $1024$ hidden neurons managed to reconstruct all the images when presented with $1/8$ of the original image, and more than half for the smallest fraction considered, $1/16$. The network with $2048$ neurons also failed to reconstruct all the images when presented with $1/16$ of the original image. However, even when presented with a portion as small as $1/16$ of the original image, the reconstruction was clear, although not perfect and hence now aove our threshold. This result is shown in Fig. 6 (left).

Surprisingly, generative PCNs trained on ImageNet correctly stored all the presented training images, and correctly reconstructed them with no visible error. This shows that PCNs can be used to store high-dimensional and high-quality images in practical setups, which can be retrieved using only a low-dimensional key, formed by a fraction of the original image. Particularly, Fig. 6 (right) shows the perfect reconstruction obtained on ImageNet when the network is presented with only $1/8$ of the original image. Further experiments on ImageNet are shown in the supplementary material, where we present multiple high-quality reconstructions.

## 5   More training data points and/or deeper PCNs

In the above experiments, a shallow architecture is able to store $100$ natural images from Tiny ImageNet, and to reconstruct them with no visible error when provided with $1/8$ of the original images. We now study how this changes when either the cardinality of the dataset is increased, or the number of provided pixels during reconstruction is updated. To study this, we have trained two models with $n = \{1024, 2048\}$ hidden units on subsets of Tiny ImageNet of cardinality $N = \{50, 100, 250, 500\}$, and reconstructed images with only $p = \{1/2, 1/4, 1/8, 1/16\}$ of the original images. According to our visualization experiments, we have noted that an image has no visible reconstruction noise when

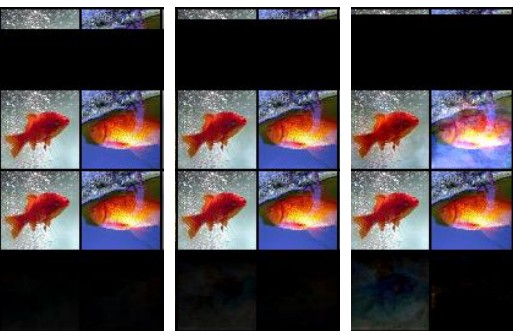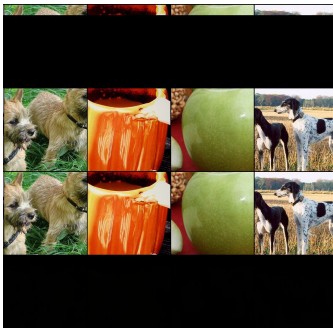

Figure 6: Partial images and their reconstructions (from top to bottom: partial image, reconstructed image, original image, and reconstruction error (difference between original and reconstruction)). Left: $64 \times 64$ Tiny ImageNet images presented with $1/4$, $1/8$, and $1/16$ of the original image, respectively, using a network of $1024$ hidden neurons. Right: $224 \times 224$ ImageNet images presented with $1/8$ of the original image, using a network with $8196$ hidden neurons.

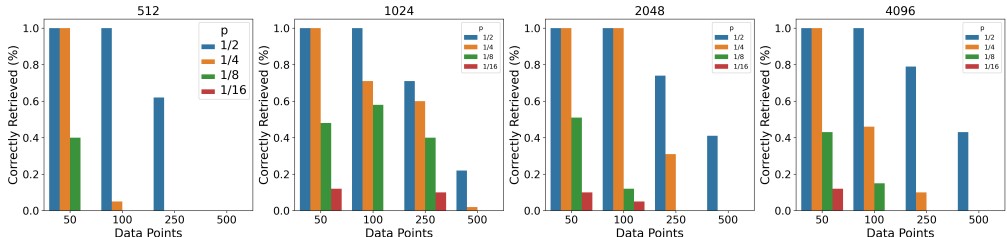

Figure 7: Percentage of perfectly retrieved images given different fractions of the original images for PCNs with $512$, $1024$, $2048$, and $4096$ hidden neurons trained on datasets of different sizes. Note that here we count as retrieved only images with non-visible errors. Most of the images that are not considered retrieved in this plot, were correctly retrieved, but only slightly noisy.

the error between the original image and the reconstruction is less than $0.001$. Hence, we consider an image to be correctly retrieved if the reconstruction error is below this threshold.

*Results:* Every network with $50$ stored memories was able to perfectly reconstruct the original image when provided with $1/4$ of it, and about half when provided with only $1/8$. However, this changes when increasing the training set, as no network trained on more than $50$ images was able to correctly reconstruct an image when provided with $1/16$ of it, besides a few cases. In terms of capacity, the more images we train on, the harder the reconstruction becomes: no training image was correctly retrieved when training on $N = 500$ images when provided with a fraction smaller than $1/2$ of the original image, regardless of the width of the network. A summary of these results is given in Fig. 7.

These experiments show the limits of shallow generative PCNs on both reconstruction capabilities and capacity. As expected and common in standard AM experiments, increasing the number of training points and reducing the number of available information for reconstruction hurts the performance. We now show how increasing the number of layers solves this capacity problem.

**Deep generative PCNs.** We now test how increasing the depth of generative PCNs increases their capacity. We then compare the results against overparametrized AEs. Particularly, we trained generative PCNs with $n = \{1024, 2048\}$ hidden neurons, depth $L = \{3, 5, 7, 10\}$ on $N = 500$ images of Tiny ImageNet. Furthermore, we reconstructed the stored images using a fraction of $p = \{1/2, 1/4\}$ of the original images. To further compare against state-of-the-art AEs, we trained equivalent AEs, which is the best-performing one according to [11]. Also, all the hyperparameters used for training are the ones reported by the authors. To make the comparison completely fair, we also assume that the correct pixels of the incomplete images are known when testing the AE. Particularly, we fixed these pixels at every iteration of the reconstruction process of the AEs. As above, we consider an image to be correctly retrieved if the reconstruction error is less than $0.001$.

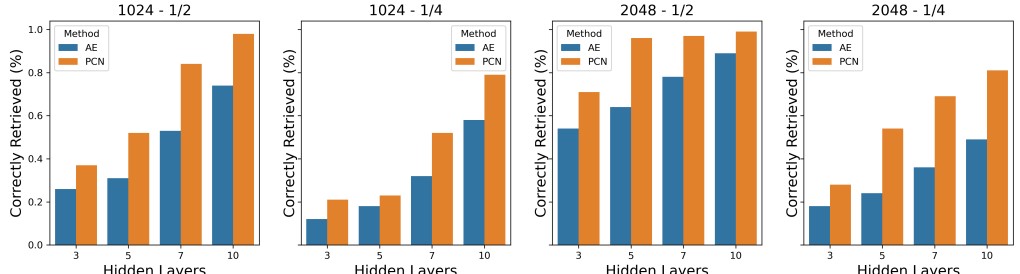

Figure 8: Percentage of correctly retrieved images as a function of the number of hidden layers, for both AEs and generative PCNs, on a dataset of $500$ images. We used networks with $n = \{1024, 2048\}$ hidden neurons, and the images are taken from Tiny ImageNet.

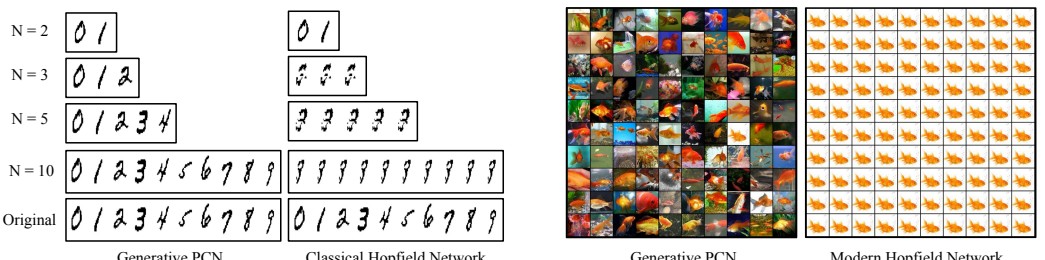

Figure 9: Left: Recovery of $2, 3, 5$, and $10$ MNIST images using a generative PCN and classical Hopfield networks [8] when presented with $1/2$ of the original image as a key. As shown, Hopfield networks only recovered the original images when $N = 2$, but otherwise always failed to retrieve them when trained on more than two images. Right: Reconstruction of $100$ Tiny ImageNet images when provided with $1/2$ of the original image, using a generative PCN and a MHN with $\beta = 2$.

*Results:* The results confirm the hypothesis: the deeper the network, the higher the capacity. Particularly, a 10-layer network was able to reconstruct more than $98\%$ (against the $72\%$ of the AE) of the images when providing half of the original pixels, and $74\%$ (against the $48\%$ of the AE) of the images when providing $1/4$ of the pixels. This shows that our model clearly outperforms state-of-the-art AEs in image reconstruction, even in the overparametrized regime. Fig. 8 summarizes these results.

## 6 Comparison with Hopfield networks

In this section, we compare against the most popular AM models, namely, classical and modern Hopfield networks (MHNs) [8, 10]. Despite their popularity, both types of Hopfield networks fail to retrieve complex data such as natural images. We now show that the retrieval process of our model is significantly better, as it always converges to a plausible solution, even when provided with a tiny amount of information, or a large amount of corruption.

**Classical Hopfield networks.** We now compare against classical Hopfield networks [8]. We have trained a Hopfield network on $N = \{2, 3, 5, 10\}$ images of the MNIST dataset, and converted every grey-scale pixel to either $0$ or $1$, as

| $N$ | $p = 1/2$ | $p = 1/4$ | $\eta = 0.2$ |
|-----|-----------|-----------|--------------|
| 50  | 5         | 4         | 4            |
| 100 | 4         | 3         | 3            |
| 250 | 1         | 1         | 9            |
| 500 | 1         | 1         | 3            |

Table 1: Number of correctly retrieved images on different tasks by a MHN trained on fractions of CIFAR10.

Hopfield networks only work on binary data. However, this model was able to retrieve the original images when presented with $1/2$ only when $N = 2$. To test generative PCNs on the same task, we used a small network with one hidden layer and 32 hidden neurons. Our model was always able to retrieve the original images. Visual results of this experiments are shown in Fig. 9 (left).

**Modern Hopfield networks.** We trained an MHN to memorize 50/100/250/500 CIFAR10 images, and then performed three different experiments. In the first two, we tried to reconstruct the original

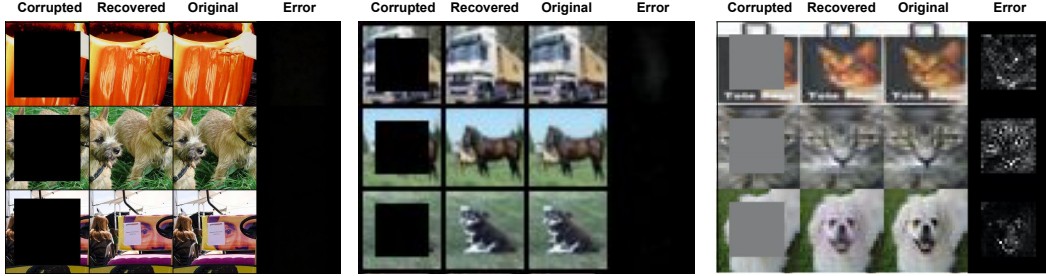

Figure 10: Left: Our reconstruction on 25 ImageNet 224×224 images and a network with 8192 hidden neurons. Centre: Our reconstruction using a generative PCN with 1024 hidden neurons on CIFAR10 32×32 images. Right: Reconstruction of a DANN on the same dataset (taken from [18]).

memories when providing $1/2$ and $1/4$ of the original image as a key. Then, we corrupted the original images by adding Gaussian noise. Despite the high capacity, MHNs do not perform well in image retrieval. Particularly, they were able to retrieve at most 9 images when presented with a corrupted image, and at most 5 when presented with an incomplete one. The results of the experiment are given in Table 1. In Fig. 9 (right), we provide a visual comparison between our method and MHNs, showing that they are only able to retrieve one image when trained on complex images taken from the Tiny Imagenet dataset. Further details about the experiments are given in the supplementary material.

**Deep associative neural networks.** To improve their performance, MHNs have been augmented with a convolutional multilayer structure for unsupervised feature detection [18]. The resulting architecture, called *deep associative neural network* (DANN), is able to perform both AM and classification tasks. We now compare the image reconstruction experiments performed in [18] with the ones performed by our generative PCN. We replicate the experiment proposed, which consists in presenting the network an partial image of the CIFAR10 dataset, where a squared patch covers the centre of each image. Note that this is not a completely fair comparison, as (1) DANNs merge AM with convolutions and unsupervised feature detection layers, and (2) it is trained on the whole dataset. Hence, comparing it against a pure AM model such as ours may be unfair. However, we believe that presenting comparisons against models that perform AM tasks such as DANNs is still interesting in understanding how our model compares against existing ones in the literature.

We first ran the same experiment on CIFAR10 as in [18]. Particularly, we used 250 images. The comparison is given in Fig. 10. Unlike the image in [18], our reconstruction shows no visible error. Furthermore, we replicated the experiment using 50 images of ImageNet, and showed that our model is again able to provide perfect reconstructions of the original images.

## 7 Biological relevance

The model described in this paper is closely related with theories of functioning of the memory system in the brain. It is typically assumed that memories are stored in a network of cortical areas organized hierarchically, where sensory information is processed and encoded by several neo-cortical regions, and then it is integrated across modalities in the hippocampus [2, 5]. It has been recently proposed that this hippocampo-neocortical system can be viewed as a PCN [6] similar to the one in this paper. In particular, Barron et al. [6] proposed that the hippocampus could be viewed as a top layer in the hierarchical PCN (which would correspond to the memory layer $L$ in our model), while neo-cortical areas would correspond to lower layers in the network. A contribution of our paper is to implement this idea and demonstrate in simulations that PCNs, which were only theoretically proposed to act as a memory system [6], can actually work and effectively store and retrieve memories.

The mechanisms of learning and retrieval proposed in the PC account of the brain memory system in [6] closely resemble those occurring in our model. Barron et al. proposed that during learning, the generative model of the hippocampus is updated until the hippocampal predictions correct the prediction errors by suppressing neo-cortical activity [6]. The training phase of our generative model closely resembles this framework, as it also accumulates prediction errors from the sensory neurons to the hidden layers and memory vector, while a continuous update of the value nodes corrects the errors in the sensory neurons. Then, the parameters of our generative model are updated until its predictions reach low/zero error in the sensory neurons. Learning in PCNs relies on local Hebbian rules, i.e., the

weights between co-active neurons are increased, and such form of synaptic plasticity has been first demonstrated experimentally in the hippocampus [19]. It has been further suggested that during the reconstructions of past memories, the hippocampus sends descending input to the neocortical neurons to reinstate activity patterns that recapitulate previous sensory experience [6]. In the reconstruction phase of our generative model, the memory vector provides descending inputs to the sensory neurons, to generate patterns that recall previous sensory experience, e.g., stored data points.

Our model captures certain high-level features of the brain memory system, but many details of the model will have to be refined to describe biological memory networks. Barron et al. [6] reviewed in detail how the architecture of a PC model could be related to the known anatomy of neo-cortical memory areas, and how different populations of neurons and connections in the model could be mapped on known groups of neurons and projections in the neo-cortex. Interesting future work is to extend the top layer of our memory network to include key features of the hippocampus, i.e., recurrent connections, known to exist in the subfield CA3, as well as an additional population of neurons in dentate gyrus and CA1, which have been proposed to play specific roles in learning new memories [20]. It is also interesting to investigate if our model can reproduce experimental data on how the hippocampus drives the reinstantiation of stored patterns in the neo-cortex during retrieval [21, 22].

## 8   Related work

Predictive coding [4] and AM models [8, 23] have rarely been related in the literature to date. However, they share multiple similarities, the main of which is that both are energy-based models that update their parameters using local information.

**Associative memories (AMs).** In computer science, AMs date back to 1961, with the introduction of the *learn matrix* [24], a hardware implementation of hetero-associative memories using ferromagnetic properties. However, the milestones of the field are Hopfield networks, presented in the early eighties in their discrete [8] and continuous [23] version. Recently, AMs have increasingly gained popularity. Particularly, a 2-layer version of Hopfield networks with polynomial activations has been shown to have a surprisingly high capacity [10], which can be increased even more with exponential activations [25, 26]. The recent focus of the machine learning community on AM models, however, does not lie in applying them directly to solve practical hetero-associative memory tasks. On the contrary, it has been shown that modern architectures are implicitly AMs. Examples of these are standard deep neural networks [27, 11] and transformers [28]. Hence, there is a growing belief that understanding AM models will help the understanding and improvements of deep learning [29, 27, 30, 31]. In fact, different classification models perform well when augmented with AM capabilities. DANNs [18], e.g., use deep belief networks to filter information, while [32] improves over standard convolutional models by adding an external memory that memorizes associations among images.

**Predictive coding (PC).** This framework includes both a biologically plausible neural architecture and a learning algorithm [4] developed to emulate learning in the visual cortex. Particularly, PC allows to describe multiple phenomena happening in the brain, using a single framework, such as free-energy minimization with local computations [33–36]. From the more practical point of view, PC can be used to train high-performing neural architectures [13, 37, 38], as it has been shown that it can exactly replicate the weight update of BP [14–16].

## 9   Conclusion

In this paper, we have shown that predictive coding (which was originally designed to simulate learning in the visual cortex) naturally implements associative memories that have a high capacity and a high retrieval accuracy and robustness, and that can be expressed using small and simple fully connected neural architectures. We have shown this empirically in a large number of experiments. Particularly, we have performed denoising and image reconstruction tasks, comparing against standard autoencoders. Furthermore, we have shown that this model is able to reconstruct with no visible error natural high-resolution images of the ImageNet dataset, when provided with a tiny fraction of them. Overall, this work strengthens the connection between the machine learning and the neuroscience community. It underlines the importance of predictive coding in both areas, both as a highly plausible algorithm to better understand how memory and prediction work in the brain, and as an approach to solve corresponding problems in machine learning and artificial intelligence.

## Acknowledgments and Disclosure of Funding

This work was supported by the Alan Turing Institute under the EPSRC grant EP/N510129/1 and by the AXA Research Fund. This work was also supported by the China Scholarship Council under the State Scholarship Fund, by the UK Medical Research Council under the grant MC_UU_00003/1, by the National Natural Science Foundation of China under the grant 61906063, by the Natural Science Foundation of Hebei Province, China, under the grant F2021202064, by the Natural Science Foundation of Tianjin City, China, under the grant 19JCQNJC00400, and by the "100 Talents Plan" of Hebei Province, China, under the grant E2019050017.

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
