# A  Detailed derivation of Eqs. (3) and (5)

## A.1  Derivation of Eq. (3)

By expanding Eq. (2) with the definition of $\varepsilon_{i,t}^l = x_{i,t}^l - \mu_{i,t}^l$, we have:

$$E_t = \sum_{l=0}^{L-1}\sum_{i=1}^{n^l}\tfrac{1}{2}(\varepsilon_{i,t}^l)^2 = \sum_{l=0}^{L-1}\sum_{i=1}^{n^l}\tfrac{1}{2}(x_{i,t}^l - \mu_{i,t}^l)^2. \tag{6}$$

Inference minimizes $E_t$ by modifying $x_{i,t}^l$ proportionally to the gradient of the objective function $E_t$. We note that each $x_{i,t}^l$ influences $E_t$ in two ways: (i) it occurs in Eq. (6) explicitly, but (ii) it also determines the values of $\mu_{k,t}^{l-1}$ via Eq. (1). Therefore, the derivative of $E_t$ over $x_{i,t}^l$ contains two terms, which are formally as follows:

$$\Delta x_{i,t}^l = -\gamma \cdot \frac{\partial E_t}{\partial x_{i,t}^l} \tag{7}$$

$$= -\gamma \cdot \left( \frac{\partial \tfrac{1}{2}(x_{i,t}^l - \mu_{i,t}^l)^2}{\partial x_{i,t}^l} + \frac{\partial \sum_{k=1}^{n^{l-1}}\tfrac{1}{2}(x_{k,t}^{l-1} - \mu_{k,t}^{l-1})^2}{\partial x_{i,t}^l} \right) \tag{8}$$

$$= \gamma \cdot \left( -(x_{i,t}^l - \mu_{i,t}^l) + f'(x_{i,t}^l)\sum_{k=1}^{n^{l-1}}(x_{k,t}^{l-1} - \mu_{k,t}^{l-1})\theta_{k,i}^l \right) \tag{9}$$

$$= \gamma \cdot \left( -\varepsilon_{i,t}^l + f'(x_{i,t}^l)\sum_{k=1}^{n^{l-1}}\varepsilon_{k,t}^{l-1}\theta_{k,i}^l \right). \tag{10}$$

Considering also the special cases of $l = L$ and $l = 0$, we obtain Eq. (3).

## A.2  Derivations of Eq. (5)

The update of weights minimizes $E_t$ by modifying $\theta_{i,j}^{l+1}$ proportionally to the gradient of the objective function $E_t$. We note that $\theta_{i,j}^{l+1}$ affects the value of the function $E_t$ of Eq. (6) by influencing $\mu_{i,t}^l$ via Eq. (1), hence, the derivative of the objective function $E_t$ over $\theta_{i,j}^{l+1}$ can be formally defined as:

$$\Delta \theta_{i,j}^{l+1} = -\alpha \cdot \partial E_t / \partial \theta_{i,j}^{l+1} \tag{11}$$

$$= -\alpha \cdot \frac{\partial \tfrac{1}{2}(x_{i,t}^l - \mu_{i,t}^l)^2}{\partial \theta_{i,j}^{l+1}} \tag{12}$$

$$= \alpha \cdot \varepsilon_{i,t}^l f(x_{j,t}^{l+1}). \tag{13}$$

# B  Further details on the experiments

Here, we provide further details about training PCNs, useful to reproduce them. Particularly, every PCN was trained using IL until convergence, we varied the following hyperparameters and reported the best result: $T \in \{1, 8, 12, 16, 24\}$ for 2-layer networks, and $T \in \{1, 8, 16, 32, 48\}$ for multilayer ones, $\gamma \in \{1, 0.5, 0.1\}$, and $\alpha \in \{0.0001, 0.00005\}$. Furthermore, we have applied a decay factor of 0.9 to $\gamma$, applied each time the energy failed to decrease.

We have used standard PyTorch initialization for the weight parameters, and initialized the value nodes to be equal to the value nodes of the previous iteration. We have always performed full batch training, and used ReLU activations in every layer. To retrieve corrupted images, we have iterated the function $F$ 30 times, using $T \in \{100, 250, 500\}$, and reported the best results. To conclude, every image has pixels normalized in $[0, 1]$ by default.

*Modern Hopfield networks (MHNs):* To perform the experiments on MHNs, we have used the official implementation, provided by the authors and available online. We now provide further details about the experiments, which are sufficient to replicate the results.

For the experiments on MHNs, the parameter $\beta$ was extensively tested, as we have used $\beta \in \{1, 2, 3, 5, 10, 100, 1000\}$, and always reported the best result. The input images are preprocessed like in all the other experiments of the paper: the images are transformed to PyTorch tensors without further modifications. For experiments comparing against classical Hopfield networks, we have converted every image to binary.

By using the above library and parameters, it is possible to replicate the conducted experiments, as no extra detail/hyperparameter is needed.

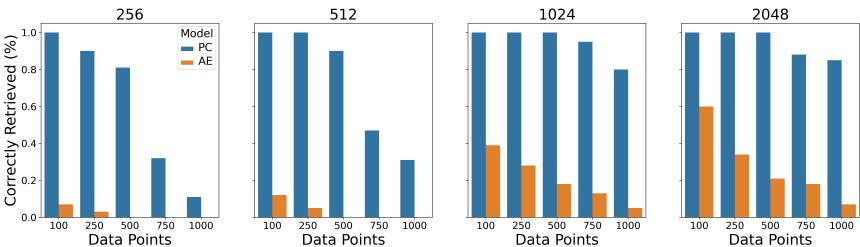

Figure 11: Percentage of correctly retrieved images by 2-layer generative PCNs (PC) and 3-layer autoencoders (AE) with hidden-layer dimensions of 256, 512, 1024, and 2048, when presented with a corrupted image with Gaussian noise of variance 0.2, trained on the Street View House Number dataset.

## C  Further results with noise of variance 0.2

In this section, we show further results of the experiments in Fig. 4. Particularly, we also include the plots for the Street View House Number dataset, excluded from the paper body due to a lack of space.

*Results:* The results, plotted in Fig. 11, are similar to the ones of CIFAR10. Again, they show the performance of our method compared to standard autoencoders.

## D  Analysis of different levels of noise

So far, we analyzed images with Gaussian noise of variance $0.2$. We now extend the analysis to different levels of corruptions. Particularly, we have used networks of width $n \in \{512, 1024, 2048\}$ trained on $N \in \{100, 250, 500\}$ natural images taken from the Tiny ImageNet dataset, and tried to retrieve them using Gaussian noise of variance $\eta \in \{0.3, 0.5, 0.7, 1.0\}$.

*Results:* The results show that our method is able to retrieve stored data points even with a larger amount of variance, although the results get worse the more we increase the level of noise. Particularly, when presented with data points with Gaussian noise with variance $0.3$, every network was able to restore at least one image, even when trained on $500$ examples. The numbers of retrieved images decreased when increasing the noise. When presented with noise with variance $\eta = 1.0$, only networks with $2048$ hidden units were able to retrieve a tiny fraction of the original data points. We report the results in Fig. 12.

## E  Plots of images with extreme levels of noise

We now plot the images generated using different levels of corruptions. Note that all the images shown here were clearly classified as *not* retrieved by the above analysis. However, we believe that an analysis of not well-retrieved images is still interesting, to understand the limitations of our method. Hence, we trained a generative PCN on $100$ images of the Tiny ImageNet dataset, and tried to reconstruct them using extreme levels of noise. Then, we printed the reconstructions.

*Results:* These representations show that generative PCNs are able to identify the original data points, even if the retrieval process was not accurate enough for the error to fall below the decided threshold $0.005$. Particularly, our model was able to identify original memories even when presented with extreme levels of noise ($\eta \in \{1.2, 1.5, 1.7, 2.0\}$), although leaving visible amount of corruption. The reconstructions are given in Fig. 13.

## F  Analysis of the retrieval function F

Here, we show visually how the function $F$ restores corrupted images. Particularly, we trained a generative PCN with $2048$ hidden neurons on $100$ images of the Tiny ImageNet dataset, and printed the reconstructions after $1$, $6$, and $11$ iterations of the retrieval function $F$.

*Results:* The reconstructions show that the first iteration is able to clear most of the noise. However, further iterations are needed to clear the remaining details, especially if the noise level is high. In fact, it can be observed that one iteration of $F$ is able to retrieve the original data point when the level

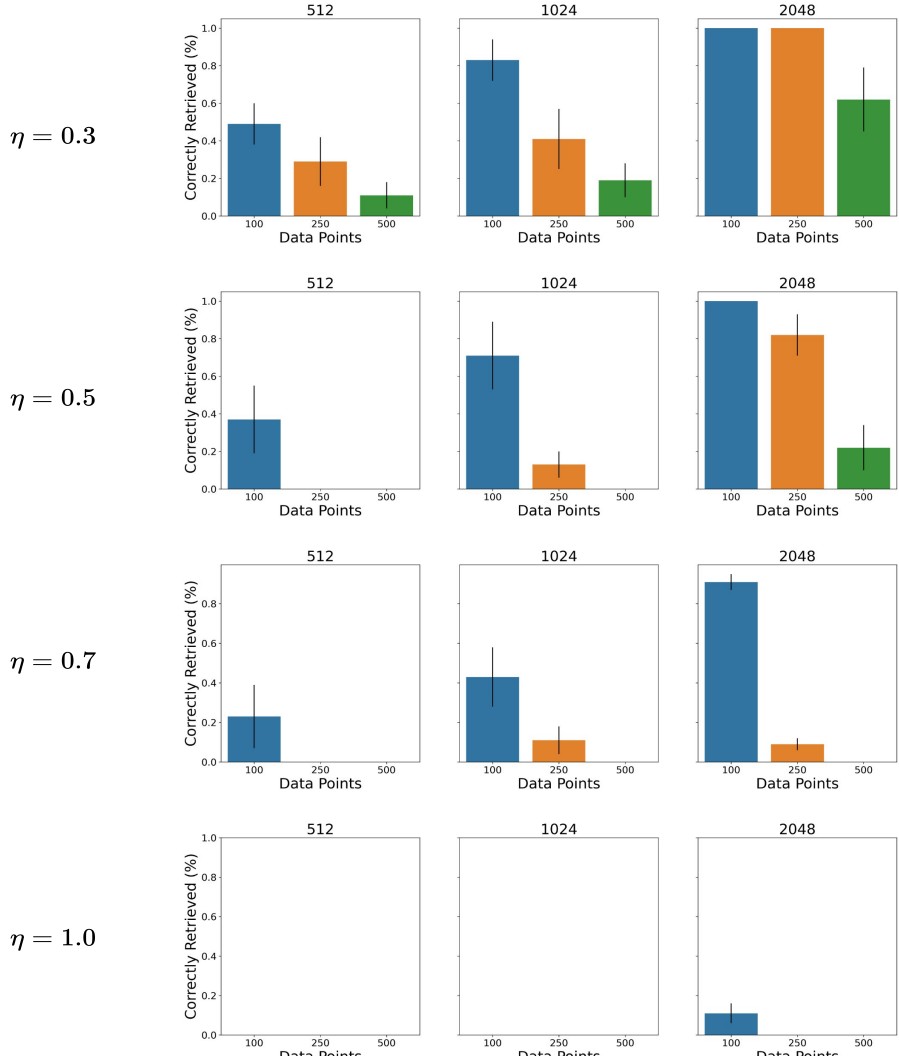

Figure 12: Percentage of correctly retrieved images by 2-layer generative PCNs (PC) with hidden-layer dimensions of 512, 1024, and 2048, when presented with a corrupted image with Gaussian noise of different variances, trained on the Tiny ImageNet dataset.

of corruption is $\eta = 0.3$, but fails when $\eta = 1.0$. This process usually converges around $m = 15$, depending on the number of iterations $T$ and the level of noise. The results are shown in Fig. 14.

## G    Reconstruction of partial images

To further show the robustness of our experiments, we replicate the experiments of Section 5.1 on the CIFAR10 dataset, training networks of different depths ($L \in \{5, 7, 10\}$) to generate 500 images, and report the results in Fig. 15. Again, we consider a data point to be perfectly reconstructed if the error between the original image and the reconstruction is below 0.001.

*Results:* The plots confirm the results stated in the body of this work: the deeper the network, the more images are perfectly reconstructed.

## H    Full-page reconstructions of ImageNet

To show the reconstruction capabilities of our method, in what follows we plot full-page reconstructions of ImageNet pictures.

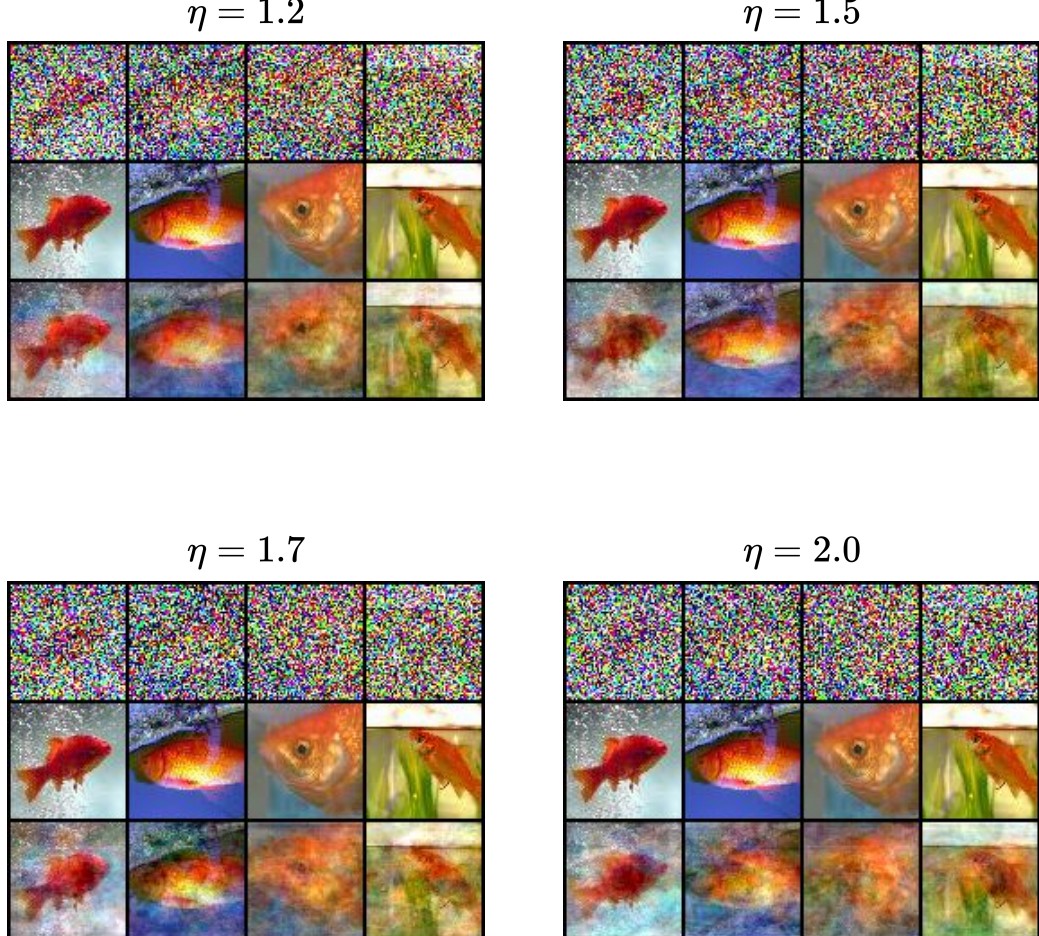

Figure 13: Analysis of the reconstruction capabilities of generative PCNs under different levels of noise $\eta \in \{1.2, 1.5, 1.7, 2.0\}$. Every plot was obtained using a 2-layer generative PCN with $2048$ hidden units, trained on $100$ images of the Tiny ImageNet dataset.

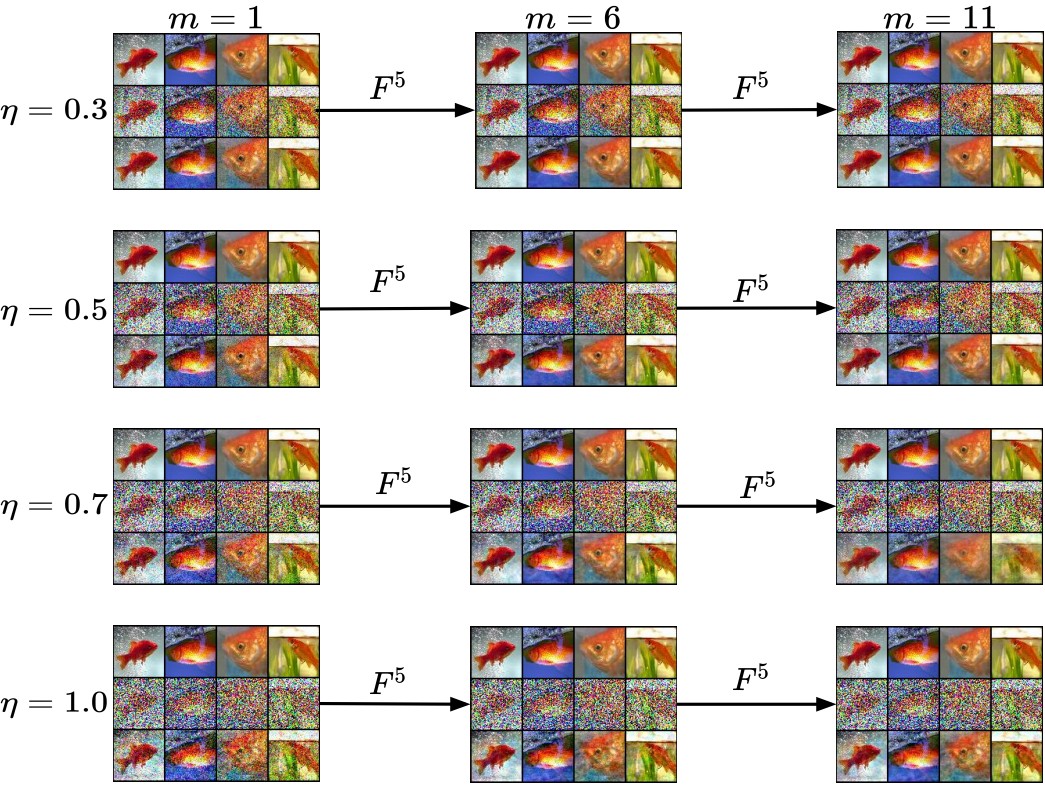

Figure 14: Analysis of the reconstruction capabilities of $10$ iterations of the function $F$ under different levels of noise $\eta \in \{0.3, 0.5, 0.7, 1.0\}$. Particularly, we plotted the results after $1$, $6$, and $11$ iterations of $F$.

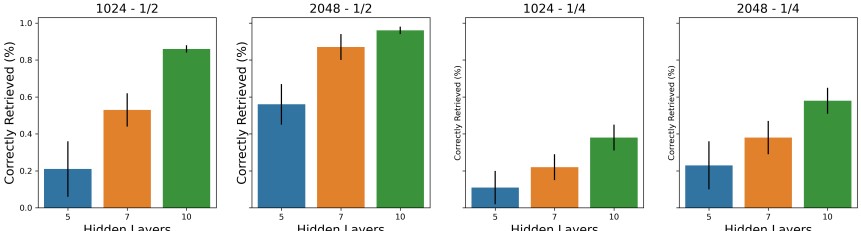

Figure 15: Percentage of correctly retrieved images as a function of the number of hidden layers, for generative PCNs on a dataset of $500$ images of the CIFAR10 dataset. We used networks with $n = \{1024, 2048\}$ hidden neurons, and tried to reconstruct them using partial images containing a fraction of $1/2$ and $1/4$ of the original images.

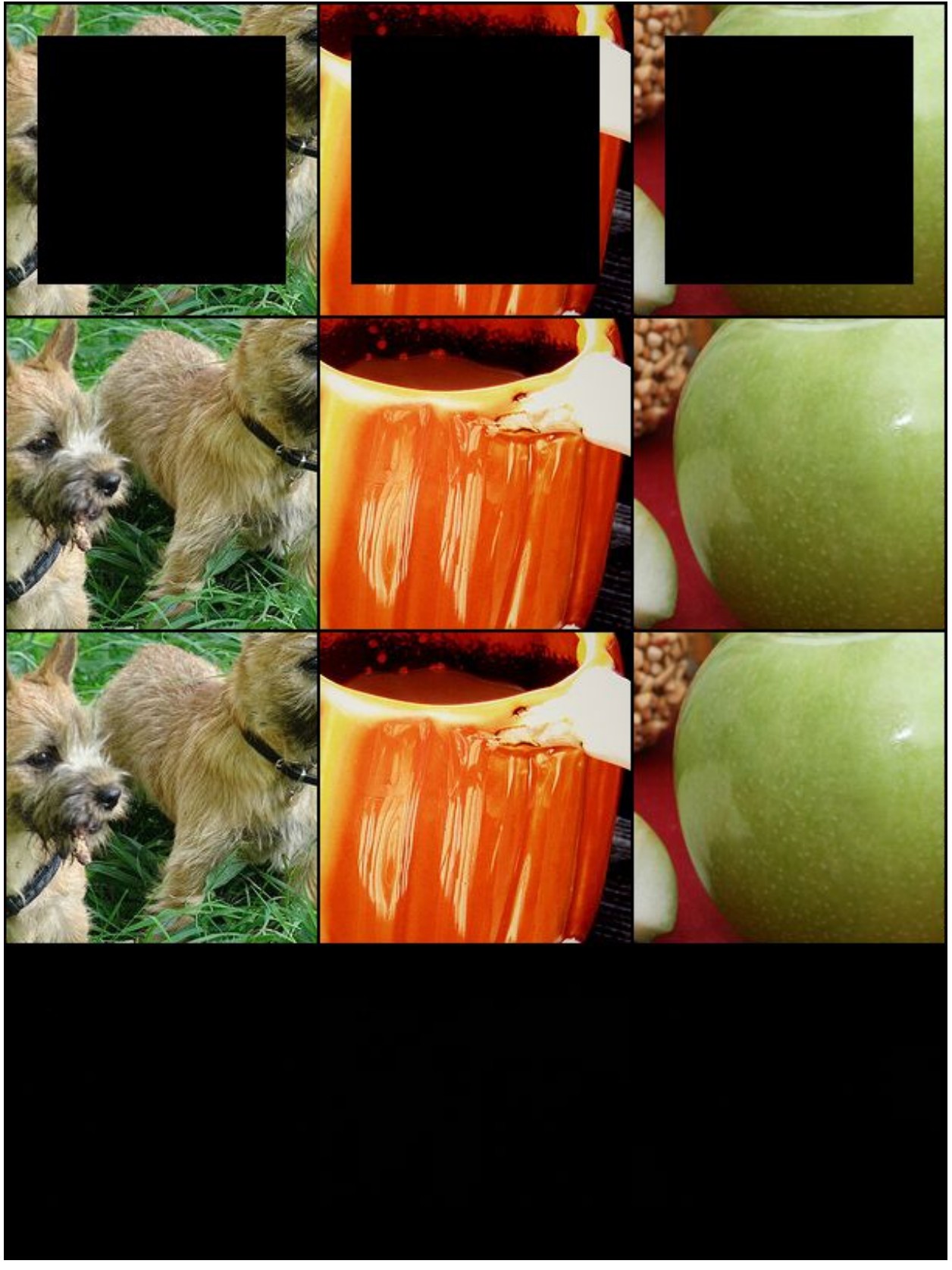

Figure 16: To show the level of accuracy of our proposed method, we represent reconstructions of ImageNet images. Here, a reconstruction when a patched image is presented to our generative PCN. From top to bottom: partial image, reconstructed image, original image, and reconstruction error (difference between original and reconstruction).

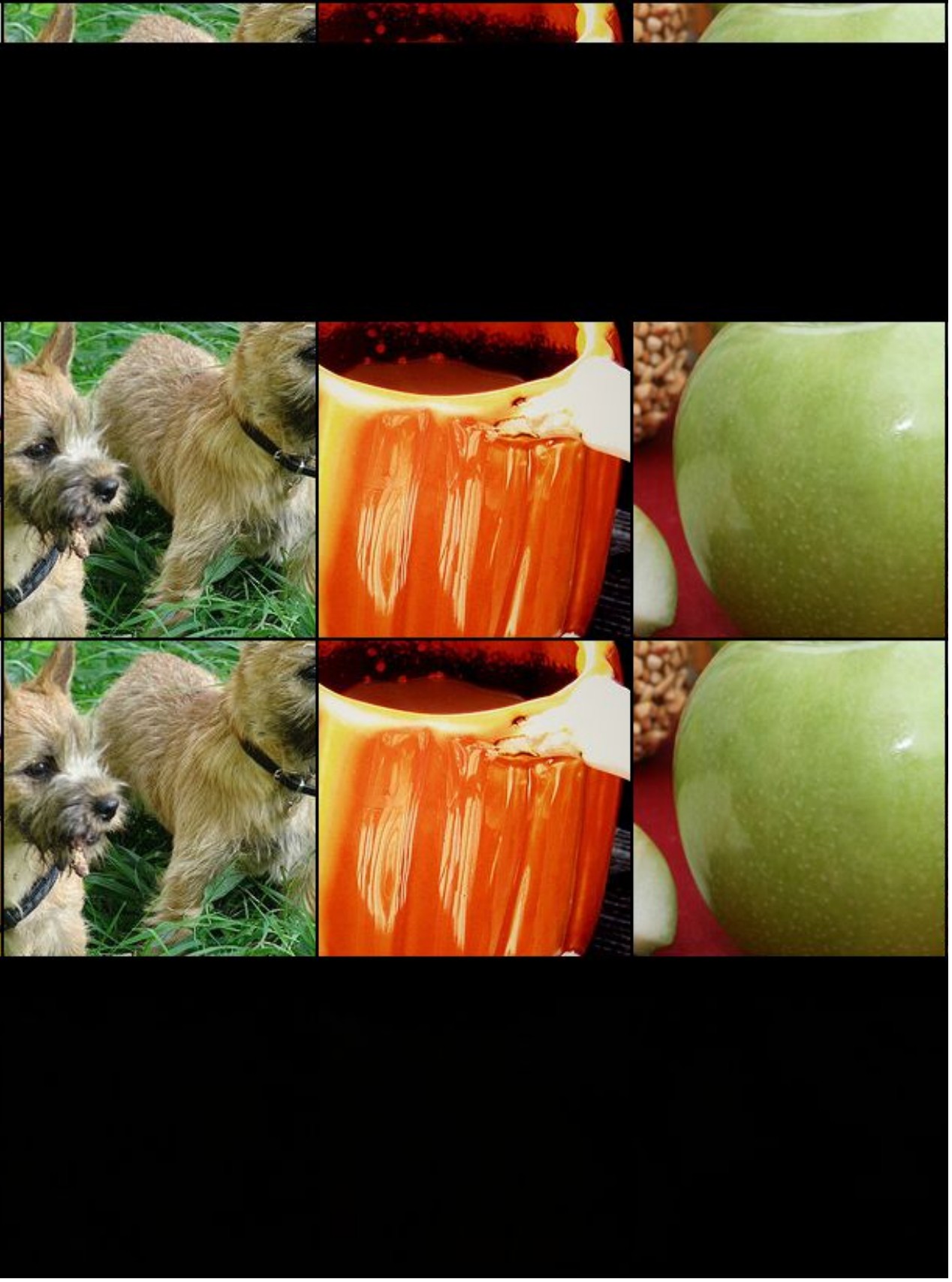

Figure 17: To show the level of accuracy of our proposed method, we represent reconstructions of ImageNet images. Here, a reconstruction when only $1/8$ of the original picture is presented to our generative PCN. From top to bottom: partial image, reconstructed image, original image, and reconstruction error (difference between original and reconstruction).