# OpenReview forum: "Associative Memories via Predictive Coding"
_NeurIPS.cc/2021/Conference — NeurIPS 2021 Poster_

### Official Review · Reviewer_SV1E · 2021-07-12

**Rating:** 5
**Confidence:** 4

**Summary:**

In this paper, the authors explore the memorization capability of a Predictive Coding (PC) model. Following training on small subsets of images taken from popular datasets (CIFAR, ImageNet), the model is probed on a variety of recall tasks that require to reconstruct images corrupted by noise or to recover entire images from partial evidence. Recall performance is compared against that of a multi-layer autoencoder trained with error backpropagation.

**Ethical Concerns:**

I do not see ethical issues related to the content of this paper.

**Limitations And Societal Impact:**

The authors could better stress the limitations of their research approach.

**Main Review:**

Originality: This research work seems mostly incremental: the authors introduce a “novel” neural model for realizing an associative memory, however the proposed architecture seems based on existing predictive coding approaches [14, 15]. Also the tasks explored are based on recently published work.

Quality: The article is well-written, and the results are clearly presented. However, I think the paper has several weaknesses that needs to be addressed:

-	A potential limitation is given by the relatively small number of data points (images) that can be accurately stored by the model: though it is interesting that the PC model can perfectly encode 500 ImageNet patterns, such number does not prove that the model can scale to realistic memory/recall scenarios.
-	I am not sure that [3] represents the state-of-the-art in (auto)associative learning models. The authors should also explore more powerful architectures, such as the PixelRNN /PixelCNN (van den Oord et al., 2016).
-	How were the training images selected? The correlation between patterns could help or hinder the storing/retrieval dynamics. The authors should investigate this issue, and possibly choose training patterns according to different correlational structure. In this respect, it could be useful to measure the “effective dimensionality” of the input, which can be defined as the number of non-zero eigenvalues in the input covariance matrix (also see Olshausen and Field, 1997).
-	It is a bit suspicious that the autoencoder with 256 hidden units is not able to learn to retrieve even just 100 images. The authors should report the trend of the loss function and carefully verify that the AE hyperparameters space was correctly explored.
-	What happens when the addition of Gaussian noise causes pixel values to go outside the range allowed? Were the values simply clipped within the [min, max] interval?
-	It would be interesting to also explore other noise types, such as salt and pepper noise, which creates less smoothed versions of the corrupted patterns. Also, in the “Retrieval from Partial Data Points” task it would be interesting to explore the possibility that the given evidence (i.e., initial pixels) might not be contiguous.
-	Figure 9: it is valuable that the authors try to compare their model with previously published results. However, from my understanding the model presented in [19] was trained to memorize (and thus reconstruct) the entire CIFAR dataset, not just a small subset of images, as in the present model. A fair comparison would thus require testing the PC model on a much more challenging setup, where the entire CIFAR needs to be memorized.
-	MNIST is mentioned among the tested datasets (Line 159), but the authors never refer to it in the remaining of the paper.

Clarity: Overall, the paper is well written and properly organized. I am not sure the sentence “In AMs, the more training points they are able to memorize, the better, and feature learning is meaningless, as there is no test set that they have to generalize to” is legitimate. In fact, associative memories with latent units could learn higher-order correlations (features) from the visible pattern to increase storing capacity through data compression.

Significance: The research topic is timely and of broad interest for both the machine learning and the computational neuroscience communities.


**Time Spent Reviewing:**

6

---

> ### Author Response · Authors · 2021-08-10
> **Author Response**
>
> We thank the reviewer for the comments.
>
> ---
> > Novelty against existing predictive coding approaches [14, 15] and recently published work.
>
> While it is true that our model is based on previous predictive coding networks, using them as a generative model capable of memorizing images and functioning as an associative memory is, to our knowledge, completely novel. In fact, we consider the simplicity of our model a strength and not a weakness:  it shows that adding complex layers/loss terms/tricks is unnecessary to store memories, as it can be done using the standard energy function of PCNs. The retrieval of partial memories is even simpler, as it suffices to fix the value nodes to be equal to the available pixels, and let the same energy function converge to the stored missing values. Furthermore, the simplicity of this model, which stores and retrieves datapoints using the same energy function, is also what allows its biological plausibility (e.g., see Friston’s free energy principle). Moreover, it demonstrates the flexibility of predictive coding networks, since the same networks can be used to perform both classification and memory retrieval based solely on what neurons are fixed during the convergence to the minimum of the energy function.
>
> ---
> > Relatively small number of data points (images) can be accurately stored by the model
>
>
> Note that we have tested image retrieval in challenging tasks, and up to a great level of precision (the ImageNet figures look identical). To achieve this, the model needs to store fewer images. However, the capacity can be significantly increased if we require the reconstruction quality to be even slightly noisy (barely visible via a human visual evaluation). We will add a discussion about it in the final version of the paper.
>
> ---
> > Why Autoencoders
>
> While autoencoders are not the SOTA models in terms of capacity (Modern Hopfield Nets can have exponential capacity), no other paper has shown reconstruction qualities and precisions close to that one. Particularly, no one has tried such reconstruction experiments from datasets more complex than CIFAR10, while the autoencoders are able to retrieve Tiny Imagenet pictures up to a great level of precision. Since our model is also able to perform high-quality image retrievals, we believe that testing against that model is coherent with the scope of our paper.
>
> To conclude, a robust comparison against Modern Hopfield Networks, which shows the significant advantages of our model against theirs when it comes to retrieval of complex memories (CIFAR10), is provided above, in an official comment of the paper visible to everyone.
>
> ---
> > Explore more powerful architectures, such as the PixelRNN /PixelCNN (van den Oord et al., 2016)
>
>
> We have used autoencoders due to their fully connected structure, which allows them to memorize single pixels without learning any pattern or use any kind of extra information (such as locality, naturally induced by convolutions). The models proposed by the reviewer, on the other hand, induce some bias on the local (PixelCNN) or sequential (PixelRNN) structure of single datapoints. This only harms the final performance of associative memory models that aim to memorize random information. Furthermore, for a fair comparison, huge convolutional models should be tested against huge convolutional PCNs.
>
> ---
> > How are training points picked?
>
>
> For every experiment, we have randomly sampled the images from the respective datasets. We have also run the very first experiments using random Gaussian vectors, and the results were unchanged.
>
> ---
> > Weak results on small autoencoder
>
>
> We have trained every autoencoder using a solid hyperparameter grid search, and the results did not change for shallow networks. The loss, however, correctly decreased and converged. In fact, a visual analysis shows that most of the training points were actually reconstructed, however, the reconstruction was noisy.  Hence, no image has passed the selected threshold. We will add a visual comparison for autoencoders as well.
>
> ---
> > What happens when the addition of Gaussian noise causes pixel values to go outside the range allowed?
>
> There is no range limit in the injected Gaussian noise
>
> ---
> > It would be interesting to also explore other noise types, such as salt and pepper noise?
>
> We have performed the proposed experiments, and the final results are (visually) very similar to gaussian noise.  We will add the performed experiments in the supplementary material.
>
> ---
> > It would be interesting to explore the possibility that the given evidence (i.e., initial pixels) might not be contiguous?
>
> It won’t make any difference, as the fully connected layers do not use local information.
>
> ---
> > MNIST is mentioned among the tested datasets
>
>
> This is a typo, we will remove it.
>
> ---
> > Comparison with DANN unfair
>
>
> Yes, DANN trains on the full dataset, but uses convolutional layers + unsupervised feature learning.  We have simply shown the comparison because of qualitative reasons. While we have stated this in the main paper, we will make it clearer.
>
> ---
> > In AMs, feature learning is not meaningless.
>
> Pure associative memory models aim to memorize random features. In fact, every capacity measure is defined on random vectors (Hopfield and Krotov’s papers). We agree with the reviewer that, when using data that is not random, i.e., of a specific kind, feature learning increases the model capacity. However, capacity in pure AM models does not take this into account. We will better discuss this point in the final version of the paper.

---

> > ### Comment · Reviewer_SV1E · 2021-08-31
> > **Post-rebuttal comments**
> >
> > I thank the authors for their thoughtful responses and clarifications. Though most of my minor concerns were addressed, I still think this work is quite incremental, and the novelty standard is not adequate for a venue like NeurIPS. I am thus not changing my overall score for the paper.

---

### Official Review · Reviewer_K5he · 2021-07-16

**Rating:** 6
**Confidence:** 4

**Summary:**

This paper is novel in that (1) it proposes a new framework to implement predictive coding -- it back-propagates the reconstruction error to optimize the activities at each hidden layer; it then uses an EM-like algorithm to update the weights of the generative model; (2) it uses this generative predictive coding framework to implement associative memory for remembering images    Using a hierarchical generative neural network to encode associative memories, traditionally realized in Hopfield net, is rather unusual and interesting.

**Ethical Concerns:**

None.

**Limitations And Societal Impact:**

Umm. Broader and societal impact statement is missing.
On the other hand, this is basic scientific work on neural network and associative memory, I don't think that there will be any negative societal impact.

**Main Review:**

They probably should have cited and compared with current literature better. They only compared the performance or capacity of their model with an auto-encoder proposed in an archived paper by Radhakrishnan et al. (2019).   Their framework has strong connections to auto-decoder, PredNet,  Variational autoencoder, iterative amortization, but none are cited.  They should compare the memory capacity and performance of their model with SOTA and standard Hopfield net.  It is not clear whether the model's memory capacity and performance as associative memory are close to state of the art. Neural field or implicit function (see recent work by Matthew Tancik et al.  NeurIPS 2020 oral) can be made to encode far more images with much higher resolution and with fewer neurons.  While there are many insufficiencies in this paper,  I  nevertheless find it creative and conceptually refreshing and might deserve a forum at NeurIPS.


**Time Spent Reviewing:**

3 hours.

---

> ### Author Response · Authors · 2021-08-10
> **Author Response**
>
> We thank the reviewer for the comments.
>
> ---
> > Comparison to generative models?
>
> Observe that the associative memory task of restoring an already seen image from an incomplete/corrupted version of it is very different from the generative task of “reasonably” completing/repairing an incomplete/corrupt version of a never seen image. For this reason, we have only compared against autoencoders, as they are the most flexible model to perform associative memory tasks. In fact, due to their fully connected structure, which allows them to memorize single pixels without learning any pattern or using any kind of extra information (such as locality, naturally induced by convolutions). Hence, we have mostly cited works that perform associative memory experiments, and avoided the literature about generative models. VAEs, for example, add probabilistic latent vectors which significantly improve the quality of the reconstruction when it comes to generating unseen images, but the randomness induced by the sampled vector harms exact images retrieval in pure associative memory experiments. The same is true for the other generative models mentioned by the reviewer. Furthermore, Tancik’s paper proposes a different way of encoding information inside a neural network. While it again greatly improves the reconstruction quality, it does not propose a new associative memory model.  However, we will discuss generative models (PredNet, VAEs, etc.) inside the related work section.
>
> ---
> > Why only Autoencoders?
>
> While autoencoders are not the SOTA models in terms of capacity, no other paper has shown reconstruction qualities and precisions close to that one. Standard Hopfield networks only encode binary information, and modern Hopfield networks were mostly tested on MNIST/simple images.  Particularly, no one has tried associative experiments from datasets more complex than CIFAR10, while the autoencoders have been shown to retrieve Tiny Imagenet pictures up to a close-to-perfect level of precision. We believe that the quality of the reconstruction of our model is very good, and we have hence tested against the best model that we found in the literature when it comes to this particular metric.
>
> > Comparison against Modern Hopfield Networks?
>
> A robust comparison against Modern Hopfield Networks, which shows the significant advantages of our model against theirs when it comes to retrieval of complex memories (CIFAR10), is provided above, in an official comment of the paper visible to everyone.

---

> > ### Comment · Reviewer_K5he · 2021-08-27
> > **Post-rebuttal comments**
> >
> > Thank you for making the comparison with modern Hopfield net.  That is a bit more reassuring.  BTW,  while there are some potential evidence at the neuronal level recording of  the visual system for predictive coding (e.g. DiCarlos's paper), they are quite tenuous.  To me, predictive coding remains, after all these years, an unverified hypothesis. In this regard, I do share reviewer EmMS and XKuQ's sentiments.  PC is inspirational, but cannot be used to justify biological relevance. Nevertheless, I still think the approach is interesting, and in fact a bit different from the traditional predictive coding framework. Joe Marino (Caltech) wrote an interesting paper connecting PC with various ML algorithms  you might find interesting.

---

> > > ### Author Response · Authors · 2021-09-03
> > > **Comments on Predictive Coding**
> > >
> > > Here are some pointers that show how relevant PC is in modern neuroscience to study and to understand the functioning of the brain.
> > >
> > > 1) The prediction on responses to prediction errors has been directly confirmed, see [Attinger17]. Here, the authors record the activity of genetically identified interneurons in visual cortex, and show that their experiments are consistent with a predictive coding interpretation of the function of visual cortex, where the balance between feedforward and top-down input underlying the computation of visuomotor mismatch is finely tuned by visuomotor experience.
> > >
> > > 2) PC is an highly influential theory in the field, as it is unique in that it is a single theory that can explain the general principle of information processing in many different brain areas, i.e., see [Friston10], where the author uses PC to explain learning in multiple domains.
> > >
> > > 3) It has been successful in explaining diverse data and phenomena in the brain, such as end-stopping and extra-classical receptive fields effects in V1, [Rao99], bistable perception [Hohwy08], illusory motion [Watanabe18], repetition-suppression [Auksztulewicz16], and attentional modulation of cortical processing [Feldman10].
> > >
> > >
> > >
> > > Hence, overall, PC is an extremely important framework used to describe learning in the neocortex. Thus, we believe that providing a computational model based on this framework, which closely resembles the storage and retrieval of memories in the hippocampus is a valid contribution in neuroscience.
> > >
> > >
> > > [Friston10] Friston, K. (2010). The free-energy principle: a unified brain theory? Nature reviews neuroscience.
> > >
> > > [Rao99] Rao, Rajesh PN, and Dana H. Ballard. "Predictive coding in the visual cortex: a functional interpretation of some extra-classical receptive-field effects." Nature neuroscience 2.1 (1999): 79-87.
> > >
> > > [Attinger17] Attinger, Alexander, Bo Wang, and Georg B. Keller. "Visuomotor coupling shapes the functional development of mouse visual cortex." Cell 169.7 (2017): 1291-1302.
> > >
> > > [Hohwy08] Hohwy, Jakob, Andreas Roepstorff, and Karl Friston. "Predictive coding explains binocular rivalry: An epistemological review." Cognition 108.3 (2008): 687-701.
> > >
> > > [Watanabe18] Watanabe, Eiji, et al. "Illusory motion reproduced by deep neural networks trained for prediction." Frontiers in psychology 9 (2018): 345.
> > >
> > > [Feldman10] Feldman, Harriet, and Karl Friston. "Attention, uncertainty, and free-energy." Frontiers in human neuroscience 4 (2010): 215.
> > >
> > > [Auksztulewicz16] Auksztulewicz, Ryszard, and Karl Friston. "Repetition suppression and its contextual determinants in predictive coding." cortex 80 (2016): 125-140.

---

### Official Review · Reviewer_XKUQ · 2021-07-16

**Rating:** 5
**Confidence:** 3

**Summary:**

The authors tested a generative predictive coding network on associative memory tasks and showed that it compares favorably against autoencoders.

**Limitations And Societal Impact:**

Nothing beyond what’s mentioned above.

**Main Review:**

Originality
I am not aware of prior work doing exactly what the authors did.

Quality
I have some concerns about the quality of this work.

Major concern:
Insufficient comparison with alternative models and overblown claims
Line 224: The author claimed that “This is, to our knowledge, the first AM model able to reconstruct ImageNet images to this level of precision”. I don’t know why the authors would claim that. Many memory networks such as those in NTM, DNC, Modern Hopfield network, etc. can store arbitrary real-valued inputs almost perfectly up to memory capacity. They should be able to store arbitrary ImageNet images as well. The authors mostly compared their networks to autoencoders, which is a weak baseline. Then the authors compared their results with one specific deep associative neural network, but even the authors admit that it’s not a fair comparison (Line 276).

Minor concerns:
In the Introduction, the authors talk about the link between associative memory models and overparametrized neural networks (Line 39-40). While True, this statement ignores the fact that many associative memory models can memorize a pattern after a single exposure, while gradient-descent trained networks mostly can’t.

In the Introduction
Saying predictive coding replicates learning in the visual cortex is overblown. It gives a sense that we already understand the learning mechanism in visual cortex, which we don’t.

This sentence is confusing/misleading.
“we have shown that this model is able to reconstruct with no visible error natural high-resolution images of the ImageNet dataset, when provided with a tiny fraction of them”
It reads as if the model can reconstruct all 1M+ ImageNet images when trained on a small number of them.


Clarity
The writing is overall understandable. But there lacks clearer discussion of the limitation of the approach. Some important details are hidden in the appendix. For example, the network has to be run for many time steps before convergence, linearly and substantially increasing the computational complexity. But I think the number of time steps used, which ranges from 12 to 48, is only reported in the appendix.

The use of predictive coding in this particular setup is not well motivated. Why should it be good here?

Significance
I have major doubts about several claims in the paper (see Quality section), so I cannot discuss the significance of these claims.


**Time Spent Reviewing:**

2

---

> ### Author Response · Authors · 2021-08-10
> **Author Response**
>
> We thank the reviewer for the comments.
>
> ---
> > Significance
>
> Observe that there are some important recent works in the intersection of deep learning and associative memories that underline the importance of associative memories for deep learning and neuroscience in general. In fact, Poggio recently argued that artificial neural networks are essentially advanced associative memory models [Poggio21]. While that work was mainly aimed at exploring the computational limits of deep neural networks (and so the upper bounds of their computational expressiveness), the idea to explore the relationship between deep neural networks and associative memories may be extremely fruitful in general: one only has to look at deep neural networks from a more abstract perspective and see them as associative memories **on the feature level**. Then, for each class of datapoints, learning in deep neural networks means (learning and) memorizing their shared feature combinations, and for each single datapoint class, we simply memorize the datapoint. This idea aligns extremely well with some recent studies on the memorization capabilities of deep neural networks. So, rather than being a niche in deep learning, associative memories may very well be at the very core of deep learning. Apart from this machine learning perspective, associative memories undoubtedly play a crucial role in the brain, and are thus highly relevant and important in neuroscience as well, especially if new (previously unknown) associative memory mechanisms are discovered that are based on important neuroscience models for learning in the brain, as in our paper.
>
> ---
> > Insufficient comparison with other models (NTM, DNC, Modern Hopfield network, etc)
>
> As for memorization-only models, observe that the associative memory task of restoring an already seen image from an incomplete/corrupted version of it is very different from the task of simply memorizing an image. Despite their large capacity, modern Hopfield networks have never been shown to be able to achieve high-quality reconstructions of pictures. In fact, they have been mostly tested on simple datasets, such as MNIST. Doing it on Imagenet pictures up to a level that makes it hard to distinguish the original and the reconstructed picture is completely a different challenge. The same applies to the other mentioned models.
>
> A robust comparison against Modern Hopfield Networks, which shows the significant advantages of our model against theirs when it comes to retrieval of complex memories (CIFAR10), is provided above, in an official comment of the paper visible to everyone.
>
> Furthermore, the mere discovery that predictive coding (as one of the most widespread neuroscience models for learning in the brain) has impressive associative memory capabilities (which were unknown to date) is a pretty exciting result on its own.
>
>
> ---
> > Single-shot learning:
>
>
> We have trained every task using full-batch training, hence all the stored memories were shown for multiple epochs, simultaneously, to the network, and learned using gradient descent.
>
> ---
> > Imagenet sentence:
>
>
> We have included that sentence, because no paper has tried AM experiments on Imagenet before: most of them focus on MNIST and some on CIFAR10, the autoencoder paper was the first one to perform them on Tiny Imagenet, and then we tried Imagenet. This is also the reason why we have only tested against autoencoders: no other approach than autoencoders has shown comparable reconstruction quality in CV tasks. Since our model is also able to perform high-quality image retrievals, we believe that testing against autoencoders is coherent with the scope of our paper.  But we will tone that sentence down in the final paper.
>
>
> ---
> > Overclaims and limitations
>
>
> We thank the author for the pointers: we will tone down the sentence about predictive coding and Imagenet. Particularly, what we meant was that PC was originally inspired from what we know about learning in the visual cortex. We will also better discuss the limitations of our approach. However, note that $T = 12/42$ is not a large number to be hidden on purpose, as it is common practice in deep multilayer PC networks to have it quite large. For example, in [Millidge20], the authors use $T=200$.
>
>
> ---
> > Why predictive coding?
>
>
> It works impressively well, which was unknown to date. Furthermore, as predictive coding is one of the most commonly accepted learning models in neuroscience, the presented approach may be closely related to how  associative memories are realized in the brain. Furthermore, there is a growing interest of both the machine learning and neuroscience communities in predictive coding and its comparisons against BP, and any work that shows interesting properties of this learning algorithm is valuable in its own right for both communities. We believe that showing that associative memory capabilities of generative PCNs are better than the ones of autoencoders using the same number of parameters is a very interesting property.
>
> Apart from the above, we have used predictive coding, because (1) its energy-based formulation naturally memorizes and retrieves images of the training set, (2) its error-driven learning framework allows (as shown) a high-quality retrieval of the memories, and (3) it is flexible in retrieving partial memories, as it suffices to fix the value nodes of the sensory layer to the available pixels, and let the energy function retrieve the others.
>
>
> [Poggio21] Poggio, Tomaso. From Associative Memories to Deep Networks. Center for Brains, Minds and Machines (CBMM), 2021.
>
> [Millidge20] Millidge, Beren, Alexander Tschantz, and Christopher L. Buckley. "Predictive coding approximates backprop along arbitrary computation graphs." arXiv preprint arXiv:2006.04182 (2020).

---

> > ### Comment · Reviewer_XKUQ · 2021-08-27
> > **Response to response**
> >
> > I thank the authors for the response and the new experiment. I had the chance to read the authors's response and the other reviews. The authors's response did clarify some concerns, but other reviews also are concerned about weak baselines. I am not fully convinced about the new Modern Hopfield Network experiment. It is hard to assess without sufficient details (how are the inputs preprocessed, hyperparameters tuned, etc). I have changed my score from 3 to 4 since the authors conducted new experiments (even though I have some concerns about it).

---

> > > ### Author Response · Authors · 2021-08-27
> > > **Response**
> > >
> > > Note that we have not implemented the experiments ourselves from scratch, but we have used the official implementation, provided by the authors:
> > >
> > > https://github.com/ml-jku/hopfield-layers
> > >
> > > Hence, we are confident that our analysis is error-free. We now provide further details about the experiments:
> > >
> > > 1) One of the qualities of modern hopfield networks, is that they only depend on one hyperparameter, $\beta$, as the hidden dimension is equivalent to the number of data points. This parameter was extensively tested, as we have used $\beta \in (1,2,3,5,10,100,1000)$, and always reported the best result.
> > >
> > > 2) To make our analysis more fair, and match the hidden dimension used by our PC model, we have also stored the images multiple times. In fact, in the reported numbers, the hidden dimension of the network is also provided: a network with $500$ hidden neurons trained on $100$ images, indicates that we have stored each image $5$ times.
> > >
> > > 3) The input images are preprocessed like in all the other experiments of the paper: the images are transformed to pytorch tensors without further modifications.
> > >
> > > By using the above library and parameters, it is possible to replicate the conducted experiments, as no extra detail/hyperparameter is needed.

---

> > > > ### Author Response · Authors · 2021-08-27
> > > > **Visual Comparison**
> > > >
> > > > In the following (anonymous) link, we also provide a visual comparison between our method and modern Hopfield networks (MHNs):
> > > >
> > > > https://anonymous.4open.science/r/Efficient-Associative-Memory-via-Predictive-Coding/MHN.png
> > > >
> > > > Left: Reconstruction of all 50 images of the CIFAR10 dataset when provided with $1/2$ of the original image. To generate the figure, we have used a generative PCN with hidden dimension $n=1024$. Right: Same task, performed using a MHN, with $\beta = 2$. Overall, our method has retrieved all the presented images, while MHN only $5$ of them: 1st, 4th, 45th, 47th, and 50th.
> > > >
> > > > We will add the above figure in the final version of the paper, together with the details needed to reproduce it.

---

> > > > > ### Comment · Reviewer_XKUQ · 2021-08-31
> > > > > **further increased score**
> > > > >
> > > > > Thanks for the explanation. I have further increased the score from 4 to 5.

---

### Official Review · Reviewer_EmMS · 2021-07-22

**Rating:** 5
**Confidence:** 4

**Summary:**

This papers uses predictive coding to train a new class of content addressable/ associative memory systems. It shows that it can perform substantially better than traditional autoencoders, with simpler architectures/fewer neurons.

**Ethical Concerns:**

No ethical concerns.

**Limitations And Societal Impact:**

No immediate negative societal impact concerns. LImitationsandcomparisontoexisticngliteraturecouldbeimprovedseemainreviesw)

**Main Review:**

Overall evaluation: The paper falls into the niche of building associative memory systems by training generative networks to the point of overfitting single training examples. While the idea is quite straightforward, it still feels odd in the sense of exploiting a pathological regime for feature learning training --which is essentially a bug from the perspective of the original training goal ---as a feature. PCNs seem to do better at this and have the advantage of naturally defining energy-based retrieval dynamics, which is a a small, but useful conceptual improvement. I am not sure how impressed one should be about the performance improvements, given that the comparisons are limited and incomplete.

Major criticisms: While the quality of some of the reconstructions is visually impressive, the numerical experiments are insufficient (see also itemized list below). For shallow networks, the comparison to VAEs only restricted to pixel noise and the partial cue scenario being evaluated mostly qualitatively. The comparison to DANNs is not convincing and purely qualitative.

Itemized questions/issues:
- Unclear: what is the dimensionality of the 'memory vector' b?
- Unclear: what is the exact functional form of the loss (not the local errors in eq2, but the actual link-to-data part of the loss)?
- Unclear: in general, how long does it take for the network to relax to a solution?
- Unclear and potentially very important: how many times is the exact same data point presented to the network (Hopfield nets are traditionally one-shot) Are training points noise-free?
-how was the threshold for correct retrieval determined? why does it change across experiments?Can you show the distribution of errors for the same comparisons?
- Related: can one see example reconstructions of the VAE for the same images? is the difference perceptually noticeable or mostly quantitative?
- More generally: why does one need a different setup for the noisy vs incomplete cue experiments, in terms of number of neurons/ number of data points?
-don't understand the statement: 'training points are stored in the memory vector' (line154) isn't b just another hidden layer with dynamic values?
-for the partial retrieval experiments: does the geometry of the mask matter? i.e. is it important that the partial cue is spatially contiguous?
- I find several statements in the intro and abstract very odd: for instance comparing energy based attractor networks with issues with single datapoint memorization in (usually feedforward) deep nets (in abstract and intro), or the statement that 'there is not test set that they need to generalize to' when there is a natural notion of the amount of noise that retrieval needs to be invariant to, which corresponds to the size of the attraction basin in a Hopfield net (see e.g. the probabilistic formalization of Hopfield network from Sommer and Dayan and any of the following work from Lengyel and collaborators).  Also 'standard neural networks' can;t generally reconstruct their inputs so their general use as content addresable memory devices is limited.  Overall, there is a sense of a lack of understanding of the conceptual context in the introductory text that negatively impacts the evaluation of the results as a whole.
-it would be help to add errorbars to the summary plots

Originality and significance: There is a growing interest in using predictive coding networks in ML context; this paper shows a novel use of PCNs for autoassociative memory. The very small N (number of items stored) limits its applicability in practice and it is unclear how the phenomenology presented generalizes to more complex architectures.  The relevance for neuroscience is minimal.

Clarity: The text for the abstract and introduction needs work. Extensive use of acronyms sometimes hurts readability.
The description of IL is somewhat hard to parse if one hasn't seen the framework before; intuitively one tends to think of an initial feedforward flow of information, followed by error computations, rather than exclusively feedback 'synapses' with dynamics that relax to a fixed point of the global energy. In this sense the parallel to the brain hinders rather than helps the exposition.

**Time Spent Reviewing:**

5

---

> ### Author Response · Authors · 2021-08-10
> **Author Response**
>
> We thank the reviewer for the comments.
>
> ---
> > The niche of building associative memory systems
>
> Observe that there are some important recent works in the intersection of deep learning and associative memories that underline the importance of associative memories for deep learning and neuroscience in general. In fact, Poggio recently argued that artificial neural networks are essentially advanced associative memory models [Poggio21]. While that work was mainly aimed at exploring the computational limits of deep neural networks (and so the upper bounds of their computational expressiveness), the idea to explore the relationship between deep neural networks and associative memories may be extremely fruitful in general: one only has to look at deep neural networks from a more abstract perspective and see them as associative memories **on the feature level**. Then, for each class of datapoints, learning in deep neural networks means (learning and) memorizing their shared feature combinations, and for each single datapoint class, we simply memorize the datapoint. This idea aligns extremely well with some recent studies on the memorization capabilities of deep neural networks. So, rather than being a niche in deep learning, associative memories may very well be at the very core of deep learning. Apart from this machine learning perspective, associative memories undoubtedly play a crucial role in the brain, and are thus highly relevant and important in neuroscience as well, especially if new (previously unknown) associative memory mechanisms are discovered that are based on important neuroscience models for learning in the brain, as in our paper.
>
>
> ---
> > Experiments
>
> While autoencoders are not the SOTA models in terms of capacity, no other paper has shown reconstruction qualities and precisions close to that one. Particularly, no one has successfully conducted such reconstruction experiments from datasets more complex than CIFAR10, while the autoencoders are able to retrieve Tiny Imagenet pictures up to a great level of precision. Since our model is also able to perform high-quality image retrieval, we believe that testing against that model is coherent with the scope of our paper.
>
> For shallow networks, it is true that the comparison against the autoencoder is purely qualitatively (pixel error between original and reconstruction). However, we believe this is a fair comparison, as (1) pixel error is directly related to the reconstruction quality, which is visibly different, and (2) it is the main metric used in the literature. We will also add a visual comparison between autoencoders and PCNs in the appendix of our work.
>
> To conclude, a robust comparison against Modern Hopfield Networks, which shows the significant advantages of our model against theirs when it comes to retrieval of complex memories, is provided above, in an official comment of the paper visible to everyone.
>
> ---
> > What is the dimensionality of the 'memory vector' b?
>
>
> The hidden dimension. We will clarify this in the final paper.
>
> ---
> > What is the exact functional form of the loss (not the local errors in Eq. 2, but the actual link-to-data part of the loss)?
>
> It is the same of Eq. 2, with the difference that the value nodes of the sensory neurons are fixed to the data and do not change during training.
>
> ---
> > In general, how long does it take for the network to relax to a solution?
>
> For a single datapoint, it takes ~15 seconds for a shallow network, and ~1 minute for a deep one (this changes with depth and width). This requires approximately 10000 iterations through the network to converge to a reconstruction error less than the threshold.
>
> ---
> > How many times is the exact same data point presented to the network (Hopfield nets are traditionally one-shot) Are training points noise-free?
>
> We have trained every network using full-batch training, so once for multiple epochs, in parallel. The number of epochs needed is specified in the appendix for each experiment. Yes, training points are noise-free.
>
> ---
> > How was the threshold for correct retrieval determined? Why does it change across experiments? Can you show the distribution of errors for the same comparisons?
>
> Yes, a distribution of the errors will be added. We have used the thresholds that provided the fairest comparison: the denoising experiments fail to have a perfect retrieval, despite the fact that most of the images look visually good. Hence, we have determined the threshold to be equal to 0.05 for the de-noising experiments. Then, with the same threshold for the retrieval of partial images, our method always successfully retrieved all the images, and so we moved to a smaller threshold, which was more informative. We will add a comparison using a threshold of 0.05 in the appendix as well.
>
> ---
> >  Can one see example reconstructions of the VAE for the same images? Is the difference perceptually noticeable or mostly quantitative?
>
> The difference is noticeable when the pixel error is significant. Furthermore, autoencoders sometimes fail to reconstruct the original image and just return random noise reconstructions (or converge to another training point). We will add extra figures (which we already have) in the appendix.
>
> ---
>
> > More generally: why does one need a different setup for the noisy vs incomplete cue experiments, in terms of number of neurons/ number of data points?
>
> Because one task is harder than the other. We believe we have used the most informative networks for each of our experiments (plus, we believe we have shown a solid number of them).
>
> ---
> > I don't understand the statement: 'training points are stored in the memory vector' (line154) isn't b just another hidden layer with dynamic values?
>
> The complete sentence reads “training points are stored in the memory vector and the weight parameters”, which states that the network memorizes training points by updating the weights and the memory vector accordingly during training. We will update this sentence to make this clearer.
>
> ---
> > Does the geometry of the mask matter?
>
> No: Our model simply memorizes single pixels, there is no learning/pattern recognition in our experiments, and using fully connected layers allows the network to not depend on local information. Hence, what matters is the number of corrected pixels, regardless of their position in the mask.
>
> ---
> > Several statements in the intro and abstract very odd:
>
> We will better address the above points of confusion. Overall, we were talking about the memorization capabilities of neural networks in general (e.g., zero loss on a training set), and hence not directly related to the ‘memorize and retrieve’ mechanism of associative memories. Generally speaking, the associative memory capabilities of a neural model are not only important on their own, i.e., to memorize and retrieve datapoints.  In fact, Poggio recently argued that artificial neural networks are essentially advanced associative memory models [Poggio21]. They are also behind the success of deep learning models. Particularly, it has been shown that deep networks trained with BP on full-batch training simply memorize the data and use them directly for prediction via a similarity function (the kernel) [Domingos20]. They are also proven to be behind the success of the attention mechanism (i.e., Hopfield networks is all you need [Ramsauer20]). For the above reasons, we believe that studying the associative memory capabilities of a neural model  also sheds light on how the model itself works.  By test set, we mean new, unseen images to reconstruct/denoise.
>
>
> ---
> > It would be help to add error bars to the summary plots?
>
>
> Will do so (they are already present in the appendix).
>
> ---
> > The relevance for neuroscience is minimal?
>
> Associative memories play a crucial role in the brain. Since PCNs are biologically plausible, the proposed associative memory mechanism could potentially be a natural way to implement short term / associative memories in cortical circuitry.
>
> Furthermore, this work can have an impact in the field of PC networks and computational neuroscience in general. Particularly, training generative networks to the point of overfitting single training examples gives a lot of information about the memorization capabilities of a network. It shows that PCNs are able to perfectly fit datasets using less parameters than standard ANNs trained with BP, which suggests that they may also need less parameters to perform classification tasks. Hence, this work is relevant for both the neuroscience and the computational neuroscience community.
>
>
> ---
> > Clarity
>
>
> We thank the reviewer for the pointers, which will be addressed in the final paper.
>
>
> [Poggio21] Poggio, Tomaso. From Associative Memories to Deep Networks. Center for Brains, Minds and Machines (CBMM), 2021.
>
> [Domigos20] Domingos, Pedro. "Every model learned by gradient descent is approximately a kernel machine." arXiv preprint arXiv:2012.00152 (2020).
>
> [Ramsauer20] Ramsauer, Hubert, et al. "Hopfield networks is all you need." arXiv preprint arXiv:2008.02217 (2020).

---

> > ### Comment · Reviewer_EmMS · 2021-08-11
> > **Post-rebuttal feedback**
> >
> > Thank you for the detailed reply. Nonetheless, the provided information does not significantly alter my overall score. The biological relevance I deem minimal because of the time scales involved - both in single retrieval episodes, where attractors seem to take a fair bit of time to converge) and at the  level of learning, where I would argue that this mechanism fundamentally does not work for episodic memory formation.

---

> > > ### Author Response · Authors · 2021-08-11
> > > **Author Response**
> > >
> > >  (1) The fact that it takes a couple of seconds of computer time to retrieve a memory does not alter its biological plausibility, as the brain performs computations much more efficiently, and hence the same process could take fractions of a second.
> > > The brain is, in fact, certainly not based on digital GPU/CPU hardware and may realize the presented algorithm in a much more natural (and efficient) way, such as using analogue relaxation dynamics to rapidly converge to the attractor state.
> > >
> > >  (2) The retrieval process takes some time, because it retrieves exact memories at the pixel level, while memory recall is not as detailed. If the inference process is stopped after a shorter period of time, it simply recalls a slightly fuzzier memory. This flexibility of variable-length computation is also highly desirable and realistic from a biological viewpoint.
> > >
> > > As neuroscientists working in this field for quite a long time, it is also rather surprising to see such criticism, since  predictive coding, which is the basis of our model, is one of the most widespread models in neuroscience; see, e.g., the following paper by H. Barron and K. Friston.
> > > https://www.sciencedirect.com/science/article/pii/S0301008220300769
> > >
> > > Specifically, we refer to Fig. 1 (C,D) in the aforementioned paper, where the authors describe the behavior of the hippocampus as a memory index and generative model. Particularly, the description of the algorithm closely resembles our proposed model. The description of the figure states the following (similarities with our generative model in **bold**):
> > >
> > > - "As a generator of predictions, or generative model, the hippocampus accumulates ascending prediction errors from neocortical neurons lower in the hierarchy"
> > >  (**Our generative model accumulates prediction errors from the sensory neurons to the hidden layers + memory vector.**)
> > >
> > > - "and responds with descending predictions to neocortex that inhibit the neocortical prediction error signals"
> > > (**Prediction errors in our sensory neurons.**)
> > >
> > > - "Left-hand panel: When the sensory input is unexpected, the resulting prediction errors are represented in the neocortical hierarchy."
> > > (**Errors in the hidden layers during the first iterations of the training process.**)
> > >
> > > - "Right-hand panel: With learning, the hippocampal generative model is updated until the hippocampal predictions ‘explain away’ prediction errors by suppressing neo-cortical activity."
> > > (**With learning, our generative model is updated until its predictions reach low/zero errors in the sensory neurons.**)
> > >
> > > - "As a memory index, the hippocampus provides descending input to the neocortex to selectively reinstate activity patterns that recapitulate previous sensory experience" (**Retrieval process: the memory vector provides descending inputs to the sensory neurons, to selectively reinstate activity patterns that recall previous sensory experience, e.g., stored datapoints.**)
> > >
> > > Hence, we believe that our model has a high biological relevance (and could have a significant impact in computational neuroscience), as it closely mimics current models of memory storage and retrieval in the hippocampus and the neocortex.
> > >
> > > Note also that our paper is **not** about episodic memory (but our approach can be extended to cover temporal/sequential data).

---

### Author Response · Authors · 2021-08-10
**Comparison with modern Hopfield networks (MHNs)**

We have performed multiple experiments using modern Hopfield networks (MHNs) [Krotov20], and the model did not perform well compared to ours.

The advantages of MHNs are (1) their quick memorization, which allows them to memorize every image in one shot, and (2) the exact retrieval of memories. However, in terms of retrieval performance, the network performs poorly compared to PCNs and autoencoders. In particular, we have trained an MHN to memorize 100/250/500 images of the CIFAR10 dataset, and then tried to reconstruct them when providing ½ and ¼ of the original image, as well as in the presence of Gaussian noise $\eta$ = 0.2. The results are summarized in the tables below (so (i) only 1 to 4, (ii) only 1 to 3, and (iii) only 2 to 9 images were successfully retrieved). To perform the experiments, we have also varied the parameter \beta (which controls capacity/performance; \beta = \{1,2,3,5,10,20,30,100\}) [Krotov16] , and we report the best results below.


Successful retrieval of missing ½

| Hidden Dim: |              500HD |     1000HD    | 2000HD  |
|--------------|:-------------------------:|:--------------:|:---------------------------:|
| 100       |       4    |      4     |       4    |
| 250 |       1      |      1        |       1       |
| 500         |      1   |      1      |        1      |

Successful retrieval of missing ¾

| Hidden Dim: |              500HD |     1000HD    | 2000HD  |
|--------------|:-------------------------:|:--------------:|:---------------------------:|
| 100       |       3    |      3     |       3    |
| 250 |       1      |      1        |       1       |
| 500         |      1   |      1      |        1      |

Successful reconstruction from Gaussian noise $\eta$ = 0.2

| Hidden Dim: |              500HD |     1000HD    | 2000HD  |
|--------------|:-------------------------:|:--------------:|:---------------------------:|
| 100       |       2    |      3     |       3    |
| 250 |       8      |      9        |       9       |
| 500         |      3   |      3     |        4      |


Visual comparison: Every correctly retrieved image is perfectly retrieved. However, every other image converges to one of the very few correctly stored images in the model. Particularly, when asked to reconstruct the missing half of an image, most of the output images are divided in two: one half, is the correct one, the second half, corresponds to a perfect reconstruction of the wrong image.

An explanation for this might be that, since MHNs try to squeeze exponentially many datapoints into a vector space, the space around each stored datapoint where we have convergence to it necessarily has to be smaller and/or overlap with the convergence space of another stored datapoint (which would explain the observed convergence to wrong datapoints).

Overall, the main difference between our model and MHNs is the retrieval process: MHNs retrieve very few (<10) datapoints of the CIFAR10 dataset perfectly. This is because of the learning procedure, which does not depend on gradient-based methods, but is exact memorization. Our model, on the other hand, uses a gradient-based method which minimizes a loss over the entire dataset. This allows every point to be classified and retrieved, with a precision that depends on the capacity (number of parameters/layers) of the model. However, the empirical results show that the precision of our model can be impressive (see experiments on ImageNet). So, the performance of the retrieval part of our model is a strong feature of our approach (after all, what is the point of memorizing if the retrieval is very bad?)


[Krotov20] Krotov, Dmitry, and John Hopfield. "Large associative memory problem in neurobiology and machine learning." arXiv preprint arXiv:2008.06996 (2020).

[Krotov16] Krotov, Dmitry, and John J. Hopfield. "Dense associative memory for pattern recognition." Advances in neural information processing systems 29 (2016): 1172-1180.

---

> ### Author Response · Authors · 2021-08-27
> **Visual comparison:**
>
> In order to further highlight the difference in performance between our method and modern hopfield networks, in the following (anonymous) link, we  provide a visual comparison between the two methods:
>
> https://anonymous.4open.science/r/Efficient-Associative-Memory-via-Predictive-Coding/MHN.png
>
> Left: Reconstruction of all 50 images of the CIFAR10 dataset when provided with $1/2$ of the original image. To generate the figure, we have used a generative PCN with hidden dimension $n=1024$. Right: Same task, performed using a MHN, with $\beta = 2$. Overall, our method has retrieved all the presented images, while MHN only $4$ of them: first, third, fourth and last.
>
> We will add the above figure in the final version of the paper, together with the details needed to reproduce it.

---

### Decision · Program_Chairs · 2021-09-27

**Decision:**

Accept (Poster)

**Comment:**

- The paper tackles an interesting direction of realizing an associative memory via predictive coding, which is quite significant considering the importance of both associative memory and predictive coding in ML and neuroscience.
- The argument about biological plausibility is controversial and rather subjective. So, I don't take this aspect into account.
- The reconstruction quality seems impressive compared to the past results.
- I think the rebuttal addressed many concerns well enough including Hopfield Net experiments, baselines, etc. The baseline seems reasonable for this specific line of work (I agree with the argument of the authors in the rebuttal.) although the experiment and discussion can be improved based on the reviewer's feedback.
- The topic matches very well to the interest of NeurIPS.
- This is a kind of very fundamental work for which we would like to see more interesting ideas even if the experiment results is not comparable to that of very well engineered large-scale systems.
- The points raised by the reviewers are important and thus should be discussed in depth in the revision. Particularly, it would make the paper stronger by discussing the limitation more clearly.
- I like the fact that the method is simple!